# Community-based surveillance of Chagas Disease: Characterization and use of Triatomine Information Posts (TIPs) in a high-risk area for triatomine reinfestation in Latin America

Valéria Carla Faria Amaral[1☯], Millena Vieira Simões de Freitas[1☯],
Gustavo Libério de Paulo[2], Silvia Ermelinda Barbosa[1], Janice Maria Borba de Souza[1‡],
Bruno Silva Amaral[1,3‡], Liléia Gonçalves Diotaiuti[1], Raquel Aparecida Ferreira[1]*

**1** Grupo de pesquisa Triatomíneos, Instituto René Rachou, Belo Horizonte, Minas Gerais, Brazil,
**2** Instituto Geociências, Universidade Federal de Minas Gerais, Belo Horizonte, Minas Gerais, Brazil,
**3** Laboratório de Interação Microrganismo-Hospedeiro, Departamento de Microbiologia, Instituto de Ciências Biológicas, Universidade Federal de Minas Gerais, Belo Horizonte, Minas Gerais, Brazil

☯ These authors contributed equally to this work.
‡ JMBdS and BSA also contributed equally to this work.
* raquel.ferreira@fiocruz.br

## Abstract

### Background

In Brazil, vector surveillance with public participation is a prioritized action in the primary prevention of Chagas Disease (CD). It is anchored in the implementation of Triatomine Information Posts (TIPs), spaces recognized by health surveillance for receiving insects suspected of being triatomines. For the first time in the scientific literature, a study is dedicated to characterizing TIPs operationally, structurally, and functionally, as well as understanding the factors that hinder their sustainability.

### Methodology

The study was conducted in one of the most vulnerable regions for CD in the Americas. Using mixed approaches applied to health, an electronic form was sent to the municipal coordinators of endemic diseases to characterize the implementation, operational status, and description of TIPs. Data from an information system were accessed to analyze their productivity. Lastly, five focus groups were conducted to capture the perception of the endemic disease coordinators regarding TIPs.

### Principal findings

100% of the municipalities did not maintain documentary records of the use and productivity of TIPs, 40% of municipalities had never implemented a TIP; of those implemented, more than 30% were deactivated, with a significant portion located in rural areas. TIPs located in areas shared with Primary Health Care facilities showed

**Data availability statement:** All data were made available in supplementary materials.

**Funding:** This work was supported by Programa de Pesquisa para o SUS, 2020 edition, project registered under number APQ-00674-20. FAPEMIG. VCFA and MVSF received research grants provided by this project. BSA received a research grant through CNPq. IRR/FIOCRUZ financed the publication of the article and infrastructure for carrying out the research. The funders had no role in study design, data collection and analysis, decision to publish, or preparation of the manuscript.

**Competing interests:** The authors have declared that no competing interests exist.

lower deactivation rates. Key factors hindering the functioning and sustainability of TIPs included the population's lack of awareness about them, the need for increased publicity of these locations, a shortage of qualified professionals, TIP distribution in hard-to-access areas, and the absence of feedback on insect examinations to residents.

## Conclusions/Significance

A scenario of heterogeneous distribution was revealed in the implementation/functioning of TIPs, as well as low public engagement and usage. There is an urgent need for health systems to be organized to regulate surveillance with public participation and to conduct awareness campaigns aimed at preventing future household reinfestations by triatomines and the resurgence of CD transmission in endemic areas.

## Author summary

In Brazil, when the community finds insects suspected of being triatomines, the insects vectors of the agent that CD, they should take them to a TIP, which may be located in public infrastructure, like schools, basic health units, or other community facilities. This study was conducted in municipalities in Minas Gerais, Brazil, which have a high risk of new cases of CD. Different data collection tools were used to characterize the TIPs, and health professionals were interviewed to describe problems related to the TIPs. No municipality recorded the delivery of insects to the TIPs. Some municipalities had never had a TIP installed; some municipalities had disused TIPs, especially in rural areas. However, the TIPs located in areas common to primary health care had a lower disuse rate. The following were mentioned: lack of knowledge among the population about TIPs; shortage of qualified professionals; distribution of TIPs in difficult-to-access areas; and lack of feedback on insect tests to residents. There is a scenario of heterogeneous distribution of implementation and operation of TIPs in the region, with low adherence and use by the population, making it urgent to carry out actions to raise awareness among the population and strengthen the TIPs.

## Introduction

Chagas Disease is considered one of the leading neglected tropical diseases [1]. Endemic in 21 Latin American countries, it affects approximately 6–8 million people worldwide [2], with around 75 million people living in areas at risk of infection by *Trypanosoma cruzi* Chagas, 1909 [2].

According to a systematic review and meta-analysis encompassing publications from 1980 to 2012, it is estimated that approximately 4.6 million people in Brazil are infected with *T. cruzi* [3]. Additionally, this country stands out MS, 2019 in the

Americas for its rich biodiversity of triatomines, with around 65 native species, 13 of which are recognized by the Brazilian Ministry of Health as having a high capacity for invading and colonizing homes and peridomestic areas, with consequent potential for transmitting *T. cruzi* to humans [4]. Thus, despite significant progress in reducing new cases of Chagas disease in the Americas, vector surveillance remains crucial for controlling transmission by native vectors in endemic areas [5]. This is a strategic action aligned with the 2030 Agenda for Sustainable Development, aiming to eliminate Chagas disease as a public health problem in endemic countries [1].

Since the 1980s, entomological control of Chagas disease in Brazil has been in the phase of vector surveillance [6], structured into active surveillance and passive surveillance or surveillance with public participation [7,8].

Active surveillance involves activities such as searching for and capturing triatomines, applying insecticides to household units, and providing educational guidance to the population, conducted by endemic disease control agents health surveillance professionals at the municipal [6]. In turn, surveillance with public participation supports vector control of Chagas disease, relying on community involvement in the surveillance process and detection of any insect suspected of being a triatomine within their household [9]. Residents are instructed to collect and deliver the suspected insect to a Triatomine Information Post (TIP), preferably the one closest to their residence [10].

Created in the mid-1970s and expanded in the 1980s, TIPs are installed by health surveillance in strategic locations within rural areas and the municipalities' headquarters in endemic regions for triatomines. These posts aim to enhance the involvement of community representatives and natural leaders in decision-making and the planning of vector control activities [9–12]. The TIPs rely on volunteer collaborators, individuals responsible for receiving insects brought by residents, who are trained for this role. After receiving the insect, the collaborator reports the notification to the municipality's endemic disease control agent. The endemic disease control agent then visits the TIP to collect the insects sent by the population, either regularly or monthly, as recommended by the Ministry of Health. The insect is subsequently taken to a state or municipal reference laboratory for taxonomic identification [12,13].

Some successful experiences involving public participation in the vector control of Chagas disease in several Latin American countries are described in the scientific literature [14–16]. Abad-Franch et al. (2011) [8] conducted an extensive systematic review on the topic, pointing out that vector surveillance involving community participation significantly increases the likelihood of detecting vector insects, as residents have a higher chance of finding triatomines in their homes on a daily basis, compared to the sporadic active search efforts performed by endemic disease control agents in Primary Healthcare Units [8]. Currently, surveillance with public participation is prioritized and recommended for monitoring household infestation by triatomines in endemic regions of the country [12], where low household vector density is observed.

It is important to contextualize that a significant part of the consolidation of the Brazilian Unified Health System (*Sistema Único de Saúde* – SUS) was due to the implementation of the Family Health Strategy in municipalities during the 1990s [17]. The Family Health Strategy is structured around a multidisciplinary team of professionals from different specialties, including the community health agent. Physically, the Family Health Strategy units are located in Primary Healthcare Units, and the community health agent is considered the mediator between the community, health professionals, and Primary Healthcare Units [18]. With this new SUS framework, it became evident, albeit informally, that many residents began delivering insects suspected of being triatomines either at the Primary Healthcare Units or directly to the community health agents [19].

The local community must be mobilized, informed, and engaged in monitoring their homes for the success and effectiveness of surveillance with popular participation. From this perspective, the community should address any insects to the designated surveillance locations within their respective countries; in Brazil, for instance, to the TIPs, as previously mentioned. However, the reality in Brazilian endemic areas reveals a scenario where most TIPs set up in the past remain underused by the population, with their operation and even location unknown and uncontrolled by health authorities. This situation directly impacts the surveillance of Chagas disease in the country, as a well-structured and operational system

involving these posts in collaboration with the community is recognized as crucial for the success of surveillance. In this regard, there is no data in the scientific literature on this issue, and for the first time, a scientific study [20,21] is dedicated to shedding light on the situation and use of TIPs in Brazil.

The study was conducted in one of the most important endemic areas for Chagas disease in Latin America, the northern region and part of the Jequitinhonha Valley in the state of Minas Gerais. Therefore, this study covers (i) the physical, structural, situational, and spatial characterization of the TIPs in the study area; (ii) analysis of the productivity and engagement of the population with the TIPs; (iii) the relevance of Family Health Strategy in the process of surveillance with popular participation and use of TIPs; and (iv) an understanding of the perceptions of health surveillance professionals regarding the operation of the TIPs in the territories, particularly the factors hindering their sustainability.

## Materials and methods

### Ethics statement

The study was submitted to and approved by the Research Ethics Committee of the René Rachou Institute under the certificate of ethical approval number 37132220.8.0000.5091.

### Background

The present study adopted a mixed-methods approach, with both objectives and approaches applied to health. The work began with an exploratory research design, encompassing both descriptive and analytical forms of research. Regarding the approach, a quantitative-qualitative data analysis was used.

Prior to data collection, awareness actions were conducted to present the proposal and invite key actors from the health management teams in the five health regions comprising 123 municipalities in Minas Gerais. In February 2021, an online meeting was held with the Regional Health Offices (RHOs) and Regional Health Superintendence (RHS) managers. Subsequently, five meetings were held with the municipalities' endemic disease coordinators. During these occasions, the proposal was presented, and goals were agreed upon with the health professionals.

### Study area

Minas Gerais is considered one of the Brazilian states with the highest prevalence of endemic Chagas disease [22]. Despite positive results from control efforts, there is still a significant number of chronic cases reported in various regions of the state [23], along with a probable underreporting of cases by public health information systems [24]. In the northeastern region of the state, the Jequitinhonha Valley is known as one of the areas with the highest incidence of the disease, with frequent presence and encounters of triatomines in households [25,26]. Similarly, the western and northern regions of the state also show significant numbers of disease cases [21].

Considering all these aspects, the study area encompasses the geographical regions mentioned above, with the study conducted in 123 municipalities in Minas Gerais that were classified as high risk for triatomine reinfestation in household units, according to a survey conducted by the State Health Department of Minas Gerais in 2006. The municipalities in this area belong to five health regions of the state. They are four RHOs—Unaí (12 municipalities), Pirapora (7 municipalities), Pedra Azul (25 municipalities), and Januária (25 municipalities)—and one RHS in Montes Claros (54 municipalities), all referred to here as health regions (Fig 1).

### Physical and situational characterization of TIPs

In the characterization of each TIP, the following parameters were described: location zone (urban or rural), type of installation site (commercial, residential, institution/Family Health Strategy) installation date, and the total number of TIPs installed per municipality. The primary methodology considered was document analysis, with municipal health

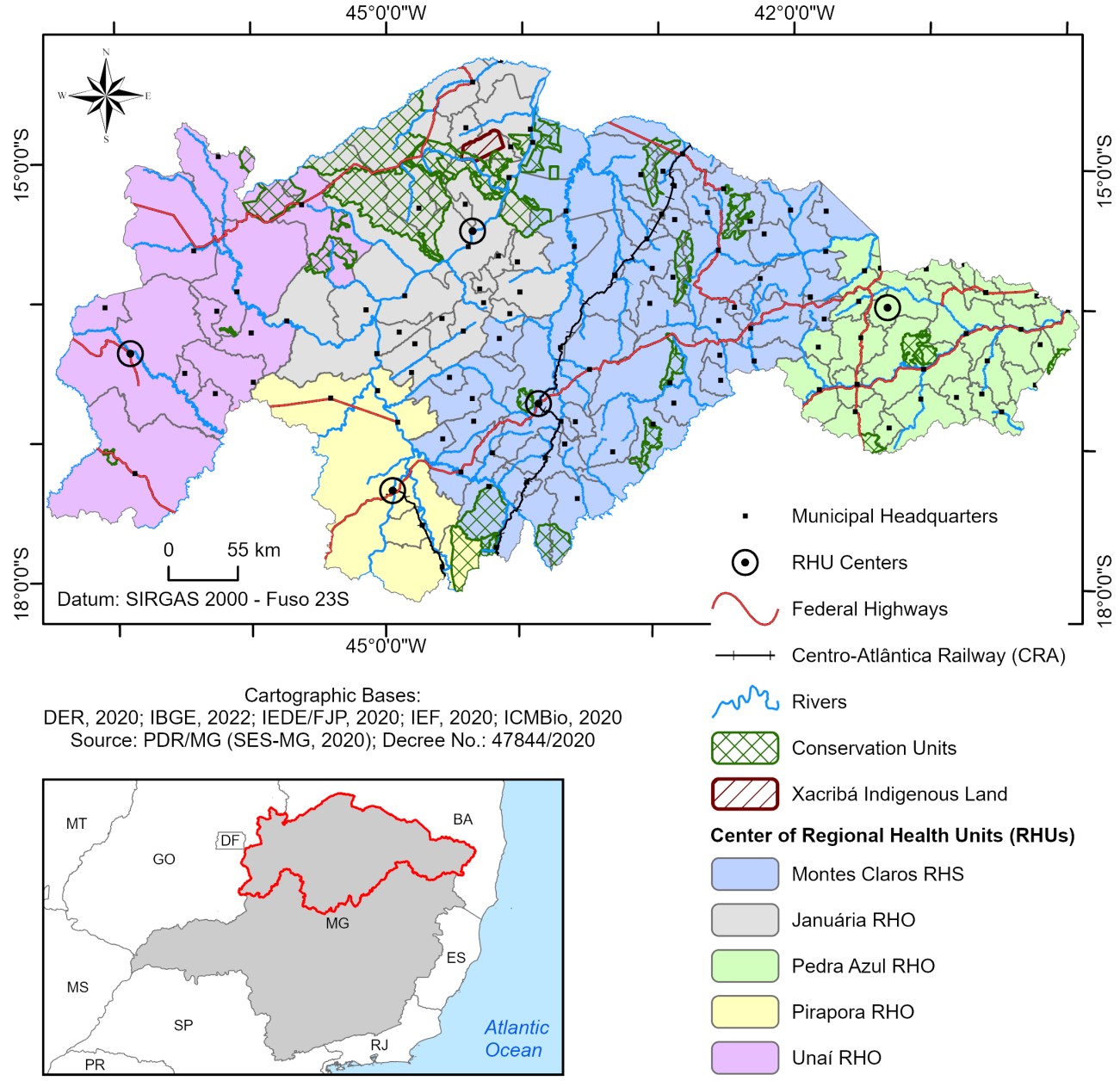

**Fig 1. Study area.** Below and to the right is the map of Minas Gerais, highlighting the study area in red outline, corresponding to the 123 municipalities belonging to the four Regional Health Offices and one Regional Health Superintendence.

managers being asked to provide copies of all standardized documents recommended by the Ministry of Health for use in TIPs [11], namely installation forms, registration books, and TIP control forms. However, it was not possible to recover these documents, as most municipalities were not using them in their routine work. As an alternative data collection method, an electronic form was created in an Excel spreadsheet containing seven questions about the TIPs installed in the municipalities. In May 2021, this form was emailed to the municipal endemic disease coordinators, requesting completion and return within 30 days. Subsequently, a spreadsheet was compiled to consolidate the

information, facilitating the creation of tables, graphs, and statistical analyses of the data (described below) using the Prisma program, Version 8.02.

As part of the TIP characterization, statistical tests were conducted (described below) to compare whether the Family Health Strategy was a variable influencing the continued operation of active TIPs in the area.

The chi-square test of independence was used for the statistical analyses, with a significance level set at 5%.

### Spatialization of the TIPs

The spatial characterization of the TIPs supported the creation of thematic maps, allowing the dynamic space-time reconstruction of the installation/deactivation of the TIP network in the study area, the location of these TIPs, the installation zone (urban or rural), and the type of installation site: commerce, residence, or institution, including information on whether they were installed in areas common to the Family Health Strategy.

The addresses of each TIP (installation and current situation in 2021) were georeferenced, and spatial data were generated using geoprocessing techniques. By cross-referencing the TIP address (location and municipality), it was possible to spatialize the data, linking it to the municipal network's digital cartographic base (shapefile) to construct the georeferenced maps. The software used for geoprocessing and the creation of thematic maps was ArcGIS version 3.00.

### Characterization of the productivity and use of TIPs

Reports of productivity from the TIPs (forms filled out by the volunteer collaborators within the TIPs or endemic disease control agent at the time of insect notification) were requested from the managers of the four RHOs and the RHS to assess the productivity and use of TIPs, that is, the number of insects received at the posts in the last five years and six months (from January 2017 to June 2021). The reports extracted from the Chagas Disease Control Program's information system (SISPCDCh) were also requested, containing data specifically related to surveillance with community participation, showing the locations and household units in the municipalities, both positive and negative for triatomines.

### Perception of health surveillance professionals about the functioning of TIPs in the territories

Municipal endemic disease coordinators are key actors in surveillance and are responsible for coordinating, supervising, and guiding control activities [12]. Therefore, these professionals hold crucial information about the overall situation of local surveillance and were strategically selected in this study to capture their perceptions on the functioning of TIPs in the territories, particularly concerning factors that facilitate or hinder their sustainability. In the absence of these actors' participation, laboratory technicians and field supervisors were consulted.

Five focus groups were conducted, one for each RHO or RHS. An invitation was sent via email to the endemic disease coordinators of all municipalities in the Pirapora RHO (seven municipalities) and Unaí (12 municipalities). For the other RHOs and the RHS with a larger number of municipalities, 14 municipalities per region were randomly selected for the invitation. The coordinators of these municipalities were then invited via email to participate in the focus groups. In the end, each focus group included six to 11 coordinators.

The focus group script contained eight questions addressing the professional's trajectory and experience, knowledge about Chagas disease vector surveillance, and TIPs. The groups were conducted from September to November 2021 through an online platform, recorded with the consent of the participants, transcribed, and submitted to content analysis according to Bardin [27] in three chronological phases. Analysis units were identified in each focus group and across them, with categories and analysis themes being named.

### Results

Five online meetings were held, and goals were set with the managers of the RHOs/RHS.

## Physical, situational, structural, and spatial characterization of TIPs

There was a 100% return of the electronic forms sent to the endemic disease coordinators of the 123 municipalities.

The implementation of TIPs in the health regions began in 1999 in the Januária RHO, accounting for 26.4% of the total TIP installations in the study area. Between 2000 and 2010, the process intensified in the Pedra Azul, Pirapora, and Unaí RHOs regions, where 39.4% of the installations occurred. In the more recent period, from 2011 to 2020, the Montes Claros RHS recorded 34.2% of TIP installations in its municipalities.

Of the 123 municipalities in the study area, 73 (59.4%) have had at least one TIP installed since the 1990s until the data collection period (Fig 2). Of the 54 municipalities in the Montes Claros RHS, 32 (59.3%) had TIPs installed, while 22 (40.7%) never had TIPs installed. Of the 25 municipalities in the Januária RHO, 17 (68%) had TIPs installed, while eight (32%) never had TIPs installed. Of the 25 municipalities in the Pedra Azul RHO, only six had TIPs installed, while 19 never had these posts installed. For this RHO, the number of municipalities with TIPs installed was lower than expected by chance (Chi-square test for independence, p = 0.0023). In turn, all 12 municipalities in the RHO Unaí, i.e., 100%, had TIPs installed. Finally, of the seven municipalities in the Pirapora RHO, six (85.7%) had TIPs installed, and only one (14.3%) did not have TIPs installed (Fig 2).

Fig 3 shows the spatial distribution of the number of TIPs installed in the 123 municipalities. In the thematic map, the shades ranging from yellow to brown indicate the number of TIPs installed in each municipality. Municipalities that have never had TIPs installed are represented in white. Municipalities in light yellow show areas with the lowest number of TIPs installed (1–7 TIPs), predominantly in the RHOs of Unaí and Pedra Azul. Darker shades, ranging from orange to brown, indicate a higher concentration of TIPs (8–22 TIPs), with the Montes Claros RHS and the Januária RHO standing out due to a significant number of TIPs installed (Fig 3).

A total of 339 TIPs were installed across 73 municipalities in the study area, with information collected for 327 of them. Table 1 presents the installation sites of these posts (commercial TIP: businesses and bars; institutional TIP: schools and Primary Healthcare Units; residential TIP: houses) and the number of posts in different health regions. Most TIPs were

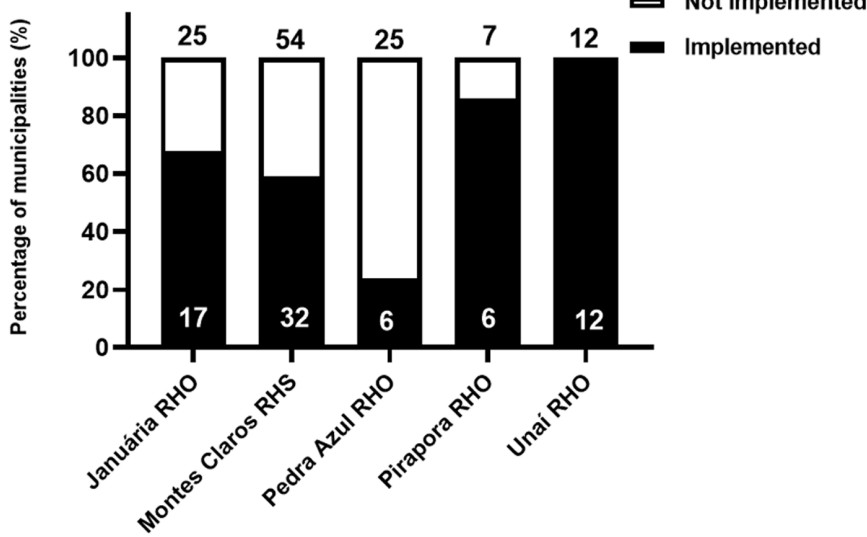

**Fig 2. Percentage of municipalities from different RHOs or the RHS that had at least one TIP installed from the 1990s until June 2021.** At the top of the bars, the absolute number of municipalities from the health regions is indicated. The absolute number of municipalities based on the evaluated situation is shown within the bars.

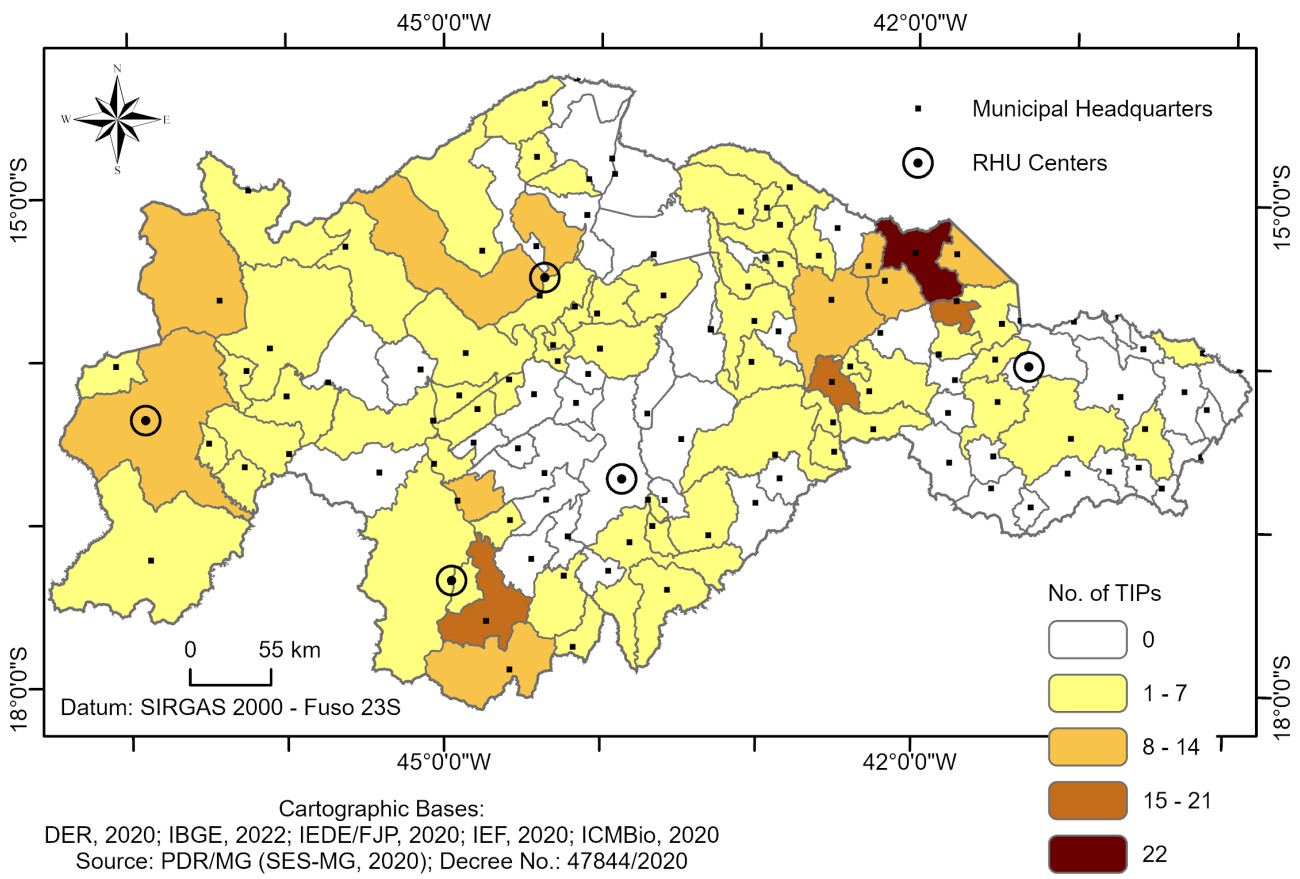

**Fig 3. Spatial distribution of TIPs installed in high-risk areas for triatomine reinfestation in household units of Minas Gerais.** The location of the municipal headquarters is indicated by black dots, and the centers of the RHUs are identified by white dots with black outlines.

**Table 1. Type of installation site for TIPs in the different health regions.**

| Health region | TIP installation site | | | Total |
|---|---|---|---|---|
| | Institutional | Residential | Commercial | |
| Montes Claros | 84 | 80 | 01 | 165 |
| Januária | 47 | 11 | 0 | 58 |
| Pedra Azul | 24 | 0 | 0 | 24 |
| Unaí | 20 | 06 | 01 | 27 |
| Pirapora | 25 | 28 | 0 | 53 |
| **Total** | **200** | **125** | **02** | **327** |

installed in health institutions (200), accounting for 61.2%, followed by 125 in residences (38.2%), and only two—0.6%—in commercial locations (Table 1).

Table 1 presents the distribution (number) and installation sites of the TIPs across the health regions. In the Montes Claros RHS, 165 TIPs were installed, with 84 (51%) in institutional locations, 80 (49%) in residences, and 1 (1%) in a commercial site. In the Januária RHO, 47 TIPs (81%) were installed in institutional locations, while 11 (19%) were in residences. In the Pedra Azul RHO, all 24 TIPs (100%) were installed in institutional locations. In the Unaí RHO, most TIPs

were also installed in institutional locations (20, 74%), followed by six (22%) in residences and one (4%) in a commercial site. Finally, in the Pirapora RHO, 28 TIPs (53%) were installed in residences, and 25 (47%) were installed in institutional locations.

The spatial distribution and number of TIPs by municipality, according to their installation sites, can be observed in S1 Fig.

Of the 339 TIPs installed in the study area, 222 (65.5%) are located in rural areas of the municipalities, while 117 (34.5%) are in urban areas (Fig 4).

Fig 5 illustrates the spatial visualization of the number of TIPs per municipality installed in the five health regions, differentiated by rural and urban zones. The number of TIPs installed in each municipality follows a similar color pattern described in Fig 3. Municipalities with one to five TIPs installed are highlighted in yellow, six to 10 in light orange, 11–15 in dark orange, and 16–19 in brown. Municipalities with no TIPs installed in that zone are shown in white.

Montes Claros RHS exhibited a higher concentration of TIPs in rural areas compared to other health regions. The RHOs of Januária and Unaí showed moderate TIP installation in rural areas. In contrast, the Pedra Azul RHO displayed fewer TIPs installed in rural zones. Lastly, the Pirapora RHO stood out for having the highest number of TIPs installed in urban areas compared to other health regions.

The analysis of the operational status of TIPs at the time of data collection revealed that out of the 339 TIPs installed in the 73 municipalities of the study area, 235 (69.3%) were still operational (referred to here as active TIPs), while 104 (30.7%) were not functioning (referred to here as deactivated TIPs) (Fig 6).

At the time of data collection, of the163 TIPs installed, 125 (76.7%) remained active and 38 (23.3%) were deactivated (Fig 6) in the municipalities of the Montes Claros RHS. In the Januária RHO, of the 60 TIPs installed, 26 (43.3%) were active, and 34 (56.7%) were deactivated. Notably, this region showed a higher proportion of deactivated TIPs than would be expected by chance (Chi-square test of independence, $p = 0$). In the Pedra Azul RHO, of the 24 TIPs installed, 15

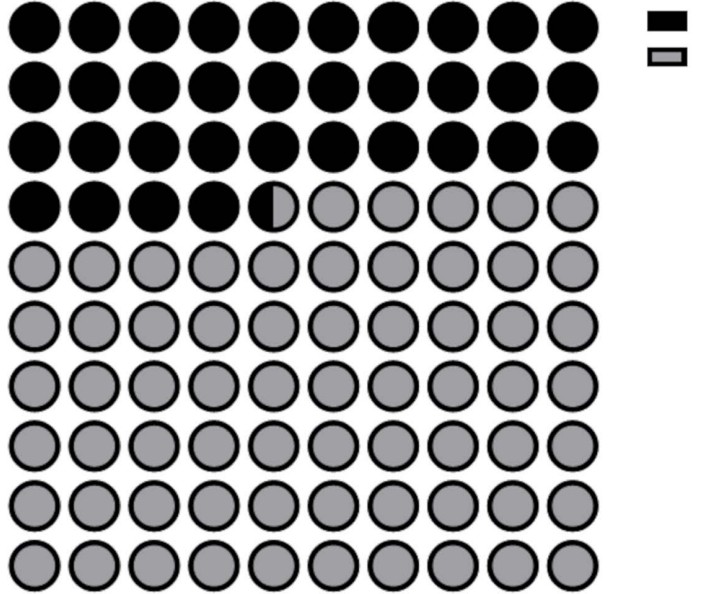

**Urban area**
**Rural area**

Overall=100

**Fig 4. Distribution (percentage) of TIPs installed in high-risk areas for triatomine reinfestation in Minas Gerais (73 municipalities) according to their location zone (urban or rural).**

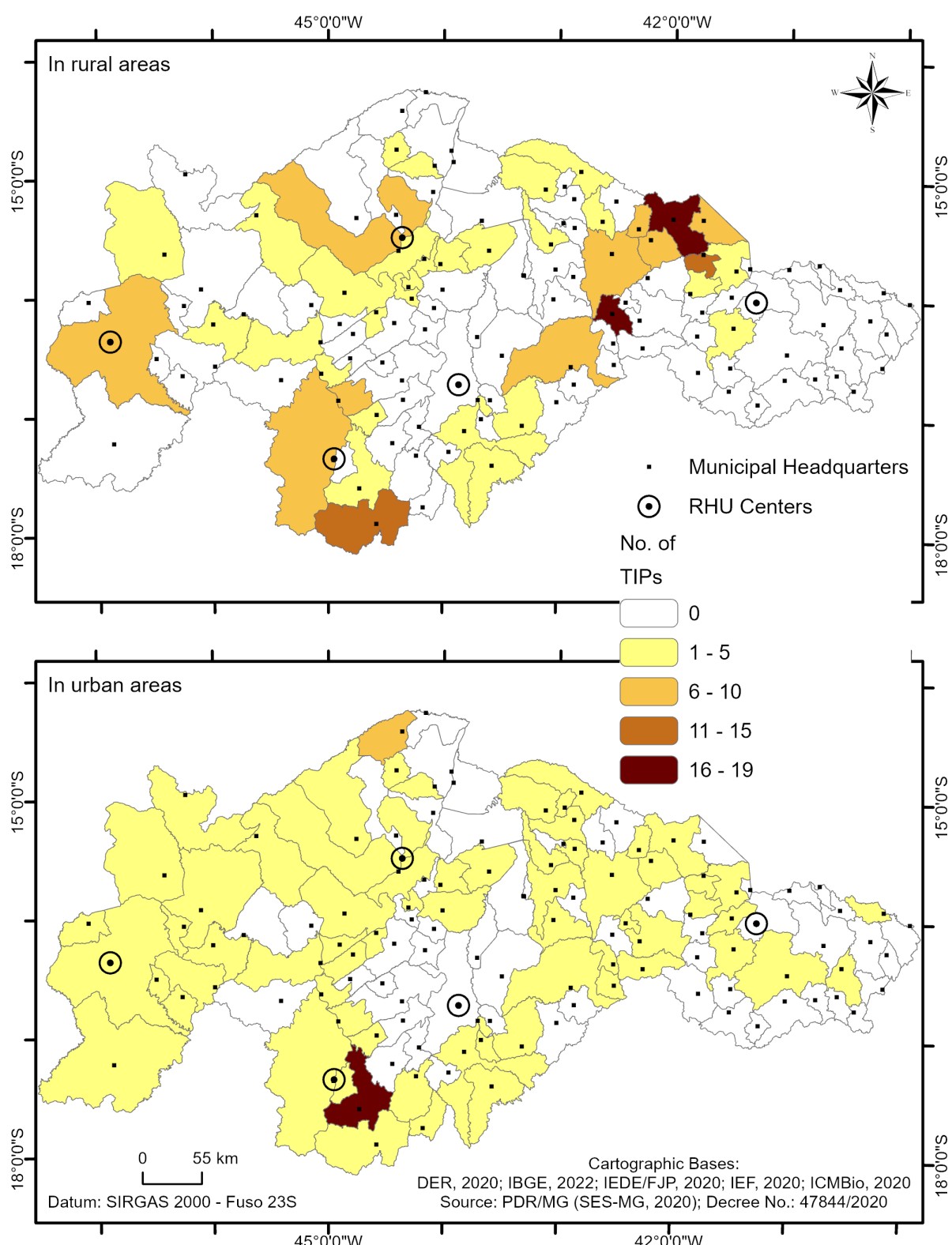

**Fig 5. Spatial distribution of TIPs installed in high-risk areas for triatomine reinfestation in Minas Gerais based on the location area.** Panel A shows the number of TIPs installed in rural areas, while map B highlights the number of TIPs installed in urban areas.

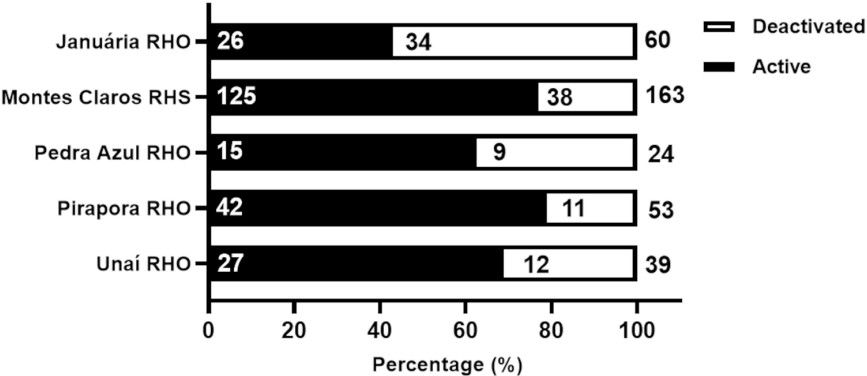

**Fig 6. Operational status (active or deactivated) of TIPs installed in areas of high risk for triatomine reinfestation in Minas Gerais (73 municipalities) according to the health region area.** Numbers outside and inside the bars indicate the total number of TIPs installed and the operational status of TIPs in June 2021.

(62.5%) were still operational, and nine (37.5%) were deactivated. In the RHO Unaí, of the 39 TIPs installed, 27 (69.2%) remained active, and 12 (30.8%) were deactivated. Finally, in the Pirapora RHO, of the 53 TIPs installed, 42 (79.2%) were active, and 11 (20.8%) were deactivated (Fig 6). This RHO had a higher rate of active TIP continuity compared to other regions.

Fig 7 shows the spatial distribution of active and deactivated TIPs by health region, supporting the findings described in Fig 6.

Distinct patterns were observed in the operational status of TIPs (active or deactivated) in relation to the geographical context—rural and urban areas in the municipalities (Fig 8). Among the 220 active TIPs, 128 (58.3%) were located in the rural areas of the municipalities, while 92 (41.7%) were in urban areas (Fig 8). Regarding the 70 deactivated TIPs, 57 (81.7%) were in rural areas, and 13 (18.3%) were in urban areas. A chi-square test of independence reveals that proportionally, there were more deactivated TIPs in rural areas than would be expected by chance (Chi-square test $p < 0.0001$). In contrast, there were more TIPs maintained in operation in urban areas than would be expected by chance.

In the context of rural areas, the operational status of the TIPs varied considerably across the health regions (Fig 9). In the municipalities of the Montes Claros RHS, of the 122 TIPs installed in rural areas, 90 (73.8%) were active, and 32 (26.2%) were deactivated. In contrast, the municipalities of the Januária RHO faced a challenging scenario, as only 12 (30%) of the 40 TIPs installed in rural areas were active, and 28 (70%) were deactivated (Fig 9). Of the five TIPs installed in rural areas in the municipalities of the Pedra Azul RHO, two (40%) remained active, and three (60%) were deactivated. In the municipalities of the RHO Unaí, of the 24 TIPs installed, 13 (54.2%) were active, and 11 (45.8%) were deactivated. Finally, of the 31 TIPs installed in rural areas of the municipalities of the Pirapora RHO, 20 (64.5%) were active, and 11 (35.5%) were deactivated. Januária and Pedra Azul RHOs had the highest proportions of TIP deactivation in rural areas.

In contrast, in the urban areas of the municipalities, the operational status of the TIPs was different from that in rural areas, with a higher percentage of active TIPs in all regions (Fig 9). Of the 41 urban TIPs installed in the Montes Claros RHS, 35 (85.4%) were active, and six (14.6%) were deactivated. In the Januária RHO, of the 20 TIPs installed in the urban area, 14 (70%) were active, and six (30%) were deactivated (Fig 9). Regarding the municipalities of the Pedra Azul RHO, of the 19 TIPs installed in urban areas, 13 (68.4%) remained active, and six (31.6%) were deactivated. Finally, in the RHO Unaí, of the 15 TIPs installed in urban areas, 14 (93.3%) were active, and only one (6.7%) was deactivated. All 22 TIPs installed in urban areas in the municipalities of the Pirapora RHO remained active (100%) (Fig 9).

We can observe the dynamics of spatialization and the number of deactivated TIPs in rural and urban areas in S2 Fig, while the dynamics and number of active TIPs in rural and urban areas is shown in S3 Fig.

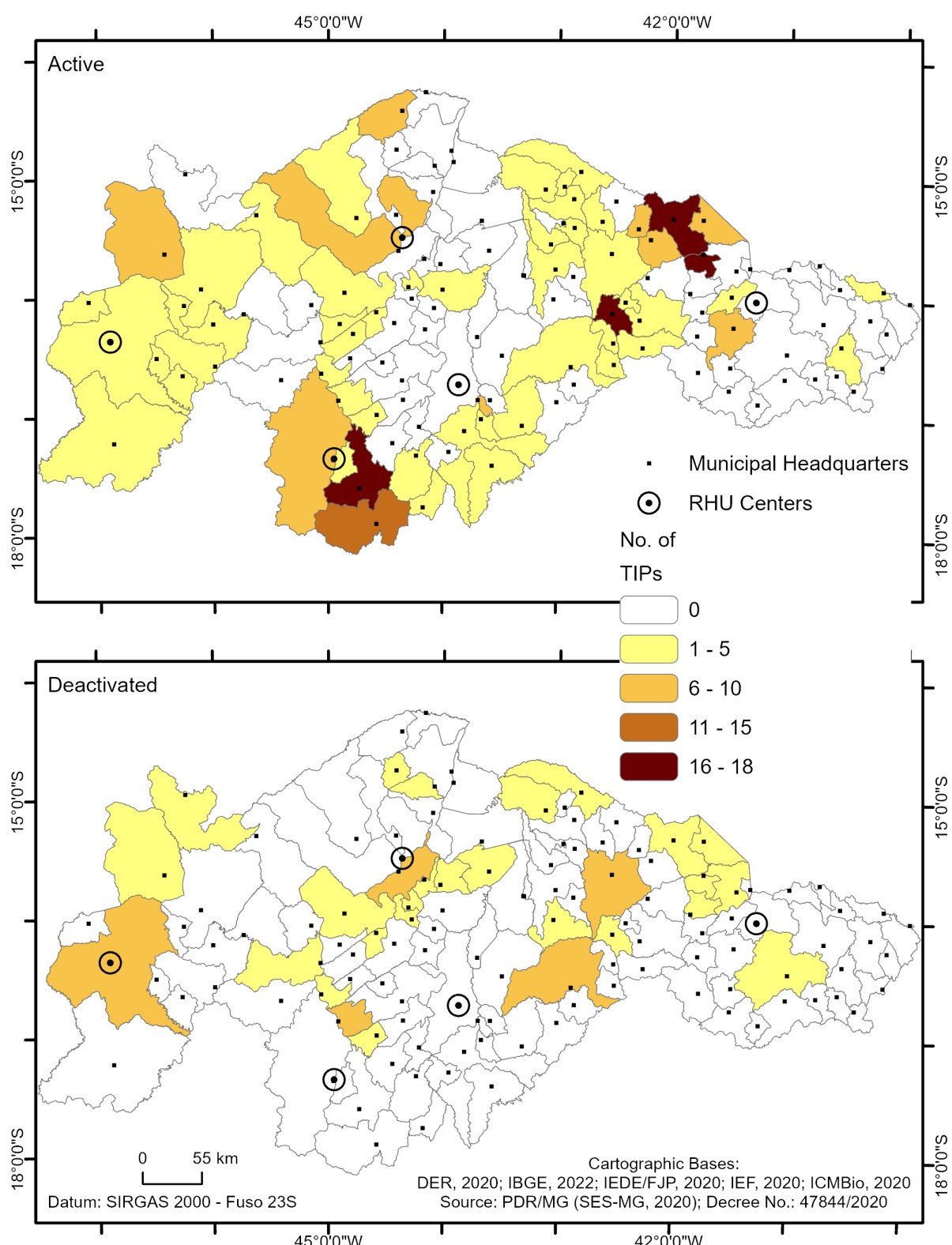

**Fig 7. Spatial distribution of TIPs in high-risk areas for triatomine reinfestation in Minas Gerais based on operational status.** Panel A shows the number of TIPs that remain active in the municipalities, while panel B shows the number of TIPs that are deactivated in the municipalities.

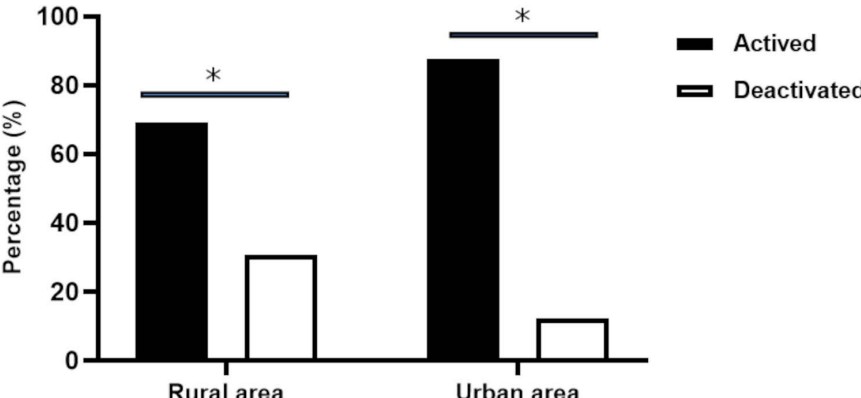

**Fig 8. Percentage of active and deactivated TIPs in relation to rural and urban areas in high-risk areas for triatomine reinfestation in Minas Gerais (73 municipalities), according to data collected in June 2021.**

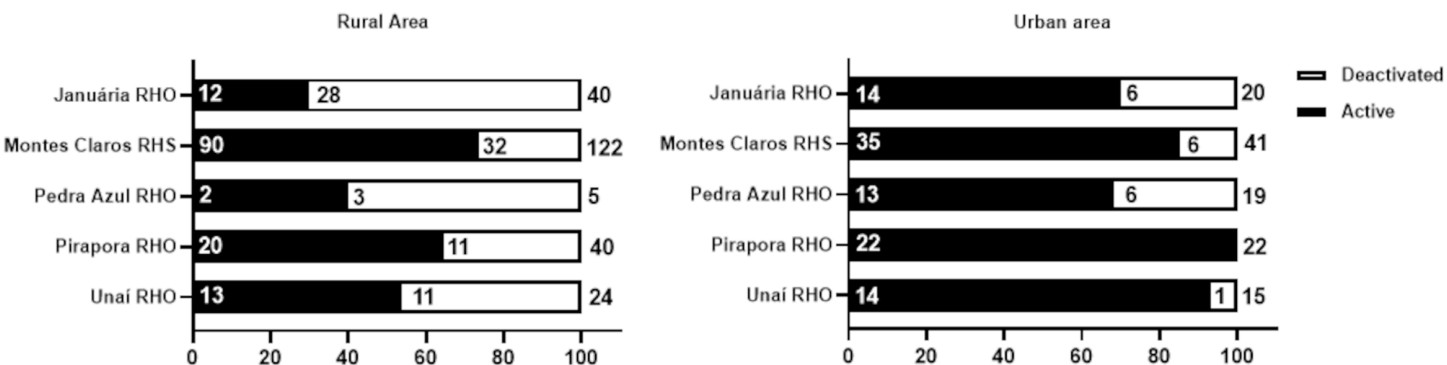

**Fig 9. Percentage of active and deactivated TIPs in relation to rural and urban zones in the high-risk area for triatomine reinfestation in Minas Gerais (73 municipalities), according to the health region.** The absolute number of TIPs in the health regions is indicated outside the bars. The absolute number of TIPs is shown inside the bars based on their operational status.

## TIPs located in common spaces of the Family Health Strategy

Of the 200 TIPs installed in public institutions, 114 were specifically located in spaces shared by the Family Health Strategy, accounting for 57% of them and representing 33.7% of all TIPs.

We analyzed whether the installation of TIPs in common spaces of Primary Health Care (PHC) influenced their operational status, that is, whether their functionality was affected when located within these services. We found that most TIPs installed in common spaces of the Family Health Strategy remained active (89; 78%), while 25 (22%) were deactivated (Table 2). Among the 225 TIPs installed outside Family Health Strategy areas, 146 remained active (64.9%), and 79 (35.1%) were deactivated. The chi-square independence test supported the hypothesis of Family Health Strategy influence on TIP functionality (p = 0.0129), demonstrating that being located in common spaces of the Family Health Strategy is a significant factor in maintaining TIPs in operation (Table 2).

Fig 10 highlights the municipalities with TIPs installed in shared locations with PHC in the study area.

Fig 11 details the operational status of the TIPs installed in areas shared with PHC across the five health regions. Of the 114 TIPs, 45 (39.5%) were located in the Montes Claros RHS, 42 (36.8%) in the Januária RHO, 11 (9.6%) in the Pedra Azul RHO, nine (7.9%) in the Unaí RHO, and seven (6.1%) in the Pirapora RHO. Regarding the operational status

**Table 2. Observed/expected values of active/deactivated TIPs based on installation location.**

| TIP location | Active | | Inactive | | Total |
|---|---|---|---|---|---|
| | Observed | Expected | Observed | Expected | |
| **TIPs in non Family Health Strategy areas** | 146 | 156 | 79 | 69 | 225 |
| **TIPs in Family Health Strategy areas** | 89 | 79 | 25 | 35 | 114 |
| **Total** | **235** | | **104** | | **339** |

of the TIPs, 34 (75.6%) were active and 11 (24.4%) were deactivated in the Montes Claros RHS. In the Januária RHO, 39 TIPs (92.9%) were active and 3 (7.1%) were deactivated. All 11 TIPs (100%) were deactivated in the Pedra Azul RHO. In the Unaí and Pirapora RHOs, all nine and seven TIPs were active, respectively (100%) (Fig 11).

The spatial distribution of TIPs installed in common areas of the Family Health Strategy and their operational status is depicted in Fig 12. Fig 13 illustrates the spatial distribution of TIPs installed in common areas of the Family Health Strategy according to their location zone. The dynamics of active and deactivated TIPs in rural areas are presented in S4 Fig, while S5 Fig shows the dynamics of active and deactivated TIPs in urban areas.

## Characterization of the use and productivity of TIPs

Only the Montes Claros RHS maintained computerized data related to the use of TIPs in the SISPCDCh, specifically data on the delivery of suspected insects brought in by the population. In the other RHOs, there was no information in the information system, nor were physical TIP records and field notes available in the surveillance service.

In the Montes Claros RHS, the number of suspected triatomine insects received at the TIPs in the municipalities varied between 2015 and December 2022 (Fig 14). The highest number of municipalities reporting the delivery of suspected insects at their TIPs occurred in 2017, with 20 municipalities (Fig 14). The second-highest percentage was observed in 2018, with 19 municipalities. In 2021, 17 municipalities reported the delivery of triatomines, followed by 2015, with 16 municipalities. The years with the lowest percentages of deliveries were 2016 and 2020, with eight and 11 municipalities receiving insects, respectively (Fig 14).

## Perception of health surveillance professionals about the functioning of TIPs in the territories

Four categories were developed based on the content analysis of the focus groups: participant profile, triatomine history of the region, control actions for Chagas disease conducted in the territories, and TIPs/surveillance with popular participation. The results related to the first three categories are presented in the Supplementary Material (S1–S3 Tables).

The profile of participants in the different focus groups (FG) according to role and sex according to role and sex is presented in the attached text in S1 Table. Similarly, the length of time the endemic disease coordinators has held their positions is described in S2 Table. The triatomine history of the region is described in S3 Table. The control actions for Chagas disease conducted in the territories are shown in S6.

Regarding the roles of the professionals participating in the focus groups, 90.9% held the position of endemic disease coordinator, while 9.1% occupied roles such as endemic disease supervisor ans data entry clerks which were classified as "Others." Of the 44 participants (S2 Table), 70.5% were men (S1 Table).

The experience time of the coordinators was clustered at two extremes. The first and largest group consisted of professionals with less than one year in the coordination position, representing 47.5% of the total. In contrast, the second largest group comprised professionals with more than 10 years in this role, accounting for 27.5% of the total (S2 Table). Additionally, 15% of the coordinators had between two and five years of experience, 5% had between five and 10 years, and only 2.5% had between one and two years of experience in the position. Finally, one coordinator did not report how long they had held the position, representing 2.5% (S2 Table).

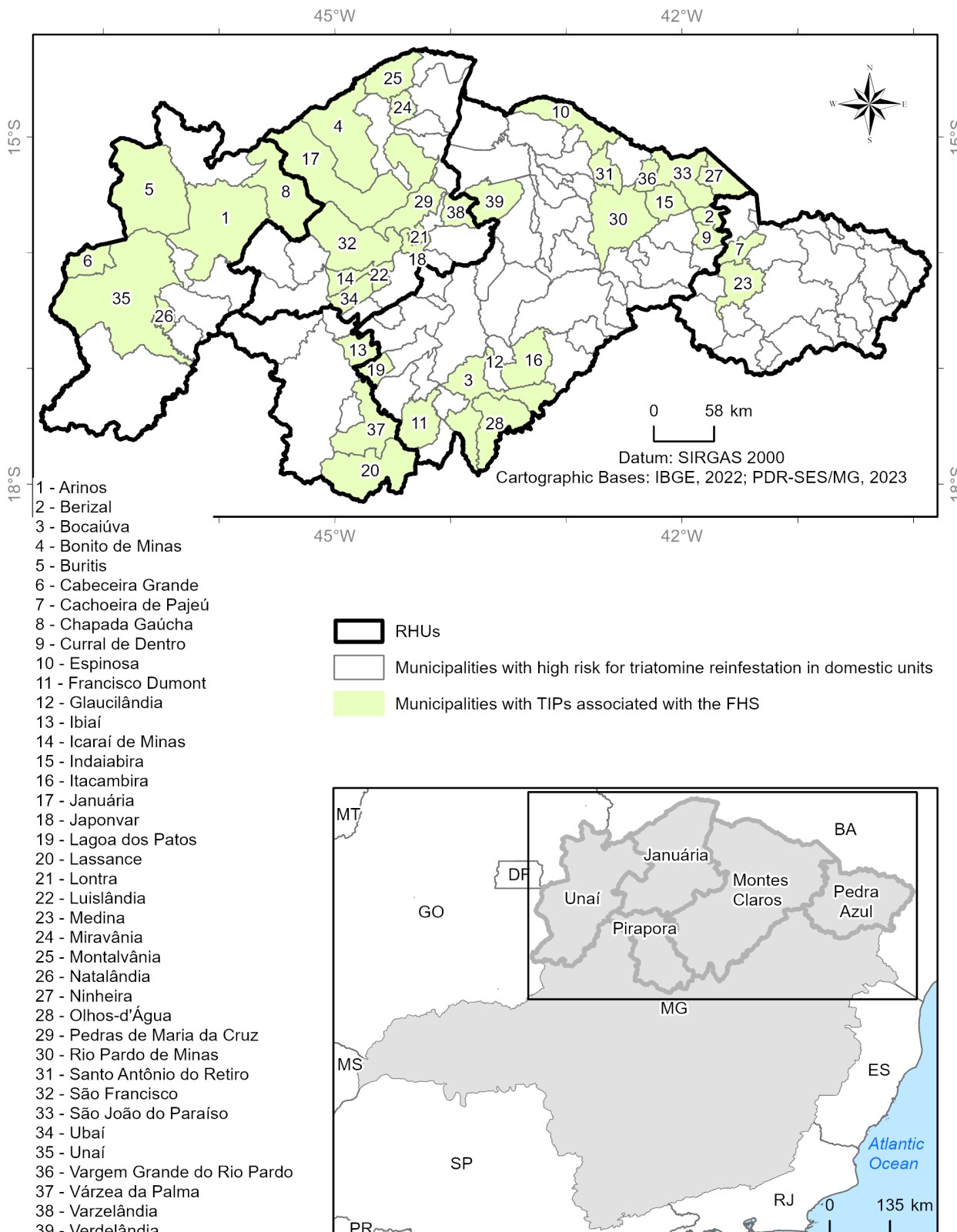

1 - Arinos
2 - Berizal
3 - Bocaiúva
4 - Bonito de Minas
5 - Buritis
6 - Cabeceira Grande
7 - Cachoeira de Pajeú
8 - Chapada Gaúcha
9 - Curral de Dentro
10 - Espinosa
11 - Francisco Dumont
12 - Glaucilândia
13 - Ibiaí
14 - Icaraí de Minas
15 - Indaiabira
16 - Itacambira
17 - Januária
18 - Japonvar
19 - Lagoa dos Patos
20 - Lassance
21 - Lontra
22 - Luislândia
23 - Medina
24 - Miravânia
25 - Montalvânia
26 - Natalândia
27 - Ninheira
28 - Olhos-d'Água
29 - Pedras de Maria da Cruz
30 - Rio Pardo de Minas
31 - Santo Antônio do Retiro
32 - São Francisco
33 - São João do Paraíso
34 - Ubaí
35 - Unaí
36 - Vargem Grande do Rio Pardo
37 - Várzea da Palma
38 - Varzelândia
39 - Verdelândia

RHUs

Municipalities with high risk for triatomine reinfestation in domestic units

Municipalities with TIPs associated with the FHS

**Fig 10. Map highlighting the 123 municipalities in Minas Gerais located in areas of high risk for domestic triatomine reinfestation.** The TIPs located in areas shared with PHC are marked in green. The map in the lower right corner shows the five health regions of the study area.

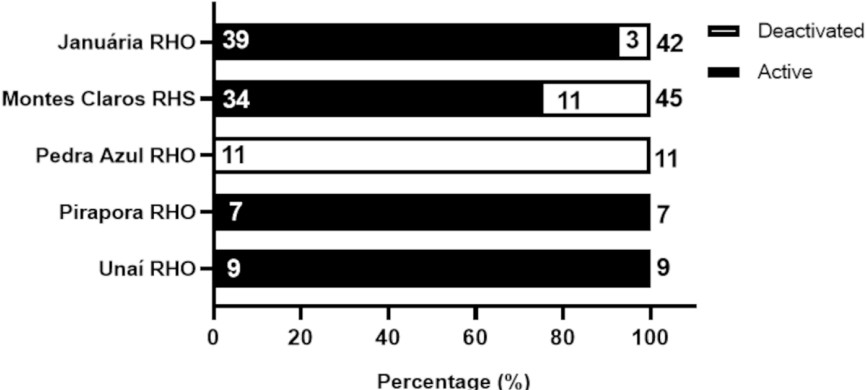

**Fig 11. Operational status of TIPs installed in common areas of PHC across different health regions with a high risk of triatomine reinfestation in Minas Gerais (39 municipalities).**

S2 Table presents the time spent in the role of endemic disease coordination by participants in the focus group. Most participants—19—had been in the position for less than one year, while 11 had more than 10 years of experience in the role.

Nearly all participants reported encountering triatomines in the territories, with the variation in responses shown in S3 Table. The majority, 38 (95%), mentioned finding triatomines, while only two (5%) did not respond. According to the participants, the most commonly found triatomine species in the municipalities was *Triatoma sordida* Stal, 1859. Other species mentioned included *Triatoma pseudomaculata* Correa & Espinola, 1964; *Triatoma vitticeps* (Stal, 1859), and *Panstrongylus megistus* Burmeister, 1835.

Regarding the Chagas disease control actions performed in the territories, 12 coordinators (30%) did not provide any input. The actions mentioned by the 28 participants are described in Fig 15. According to eight participants, no vector control actions for Chagas disease were being performed in their territories at the time of data collection. A range of different action combinations was cited by the participants, with the most commonly mentioned being TIPs + active search for triatomines in residents' homes by the EDCAs (endemic disease control agents) + spraying homes with insecticides (Fig 15). According to the participants, TIPs were prioritized as a control action for Chagas disease in 25 municipalities, or 62.5% (Fig 15). Meanwhile, the active search for triatomines was performed in at least 10 municipalities (25%).

Regarding the participants' understanding of TIPs, some statements revealed a lack of awareness about their operation and existence (first excerpt below). Additionally, comments indicated that this lack of knowledge extended to the local residents (second excerpt).

*"[...] So, since I started in this service, we knew nothing about TIP. So, it could be that it existed, but it wasn't passed on to us./... And that paperwork about Chagas, when I started, it was already gone, so if it existed, it was lost, and now it's hard for us to... get it again. [...]"*(FG 2/ Participant 2.8)

*"[...] Regarding TIP, they **(the residents)** don't even know what TIP is, but they do know that when they find a triatomine, they say: 'I'll take it to SUCAM (**surveillance**) [...]'"* (FG 2/ Participant 2.4). *Words in bold were added by us to the participant's statement.*

Regarding the operation and activity of TIPs, participants shared insights on the activity and inactivity of the posts in the territories. Some also highlighted the need for new posts and the expansion of the TIP network in the municipalities, as noted in the excerpts below:

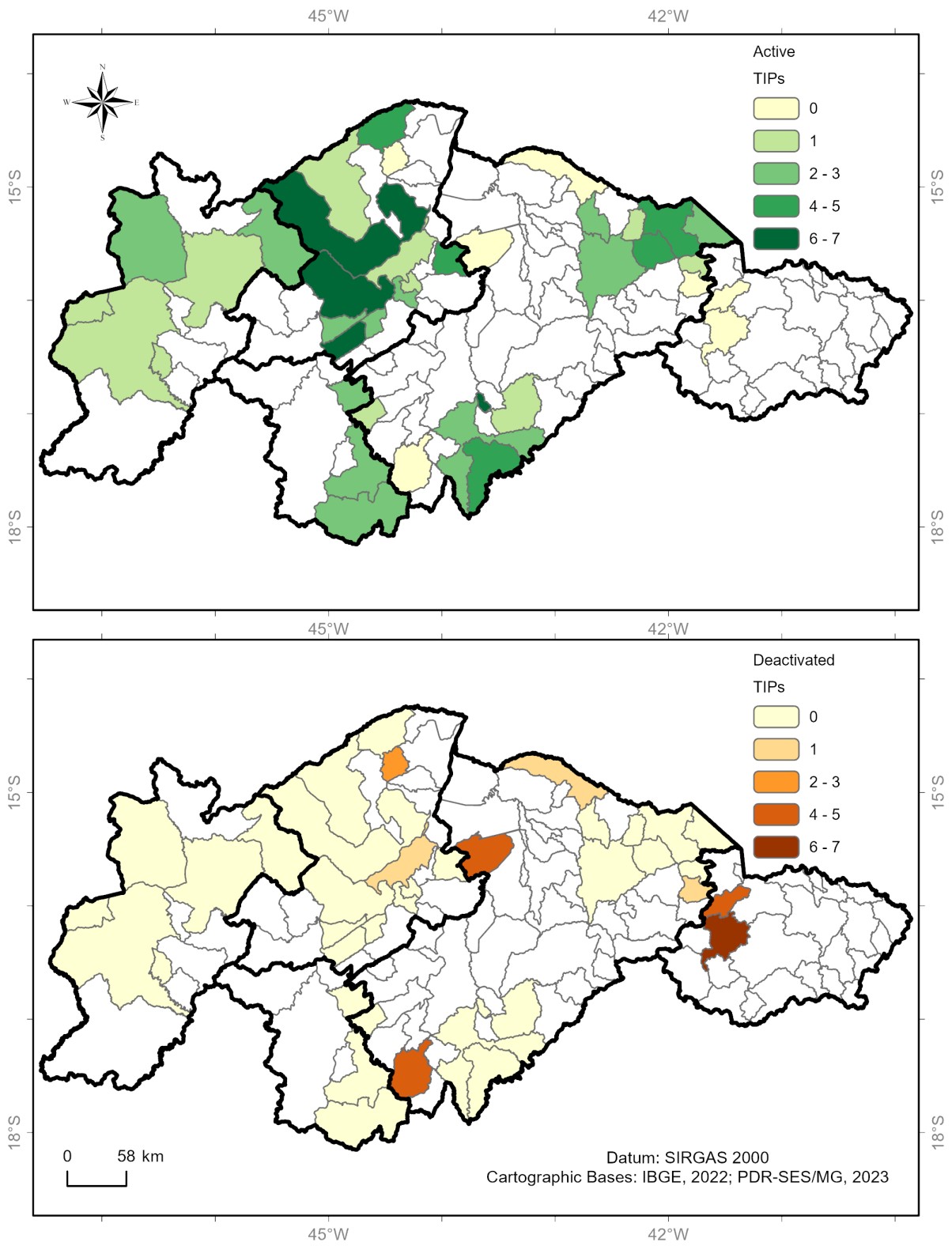

**Fig 12. Spatial distribution of TIPs located in common areas to the Family Health Strategy in areas of high risk for triatomine reinfestation in Minas Gerais based on their operational status.** Panel A shows the number of TIPs that remain active in the municipalities, and Panel B shows the number of TIPs that remain deactivated in the municipalities.

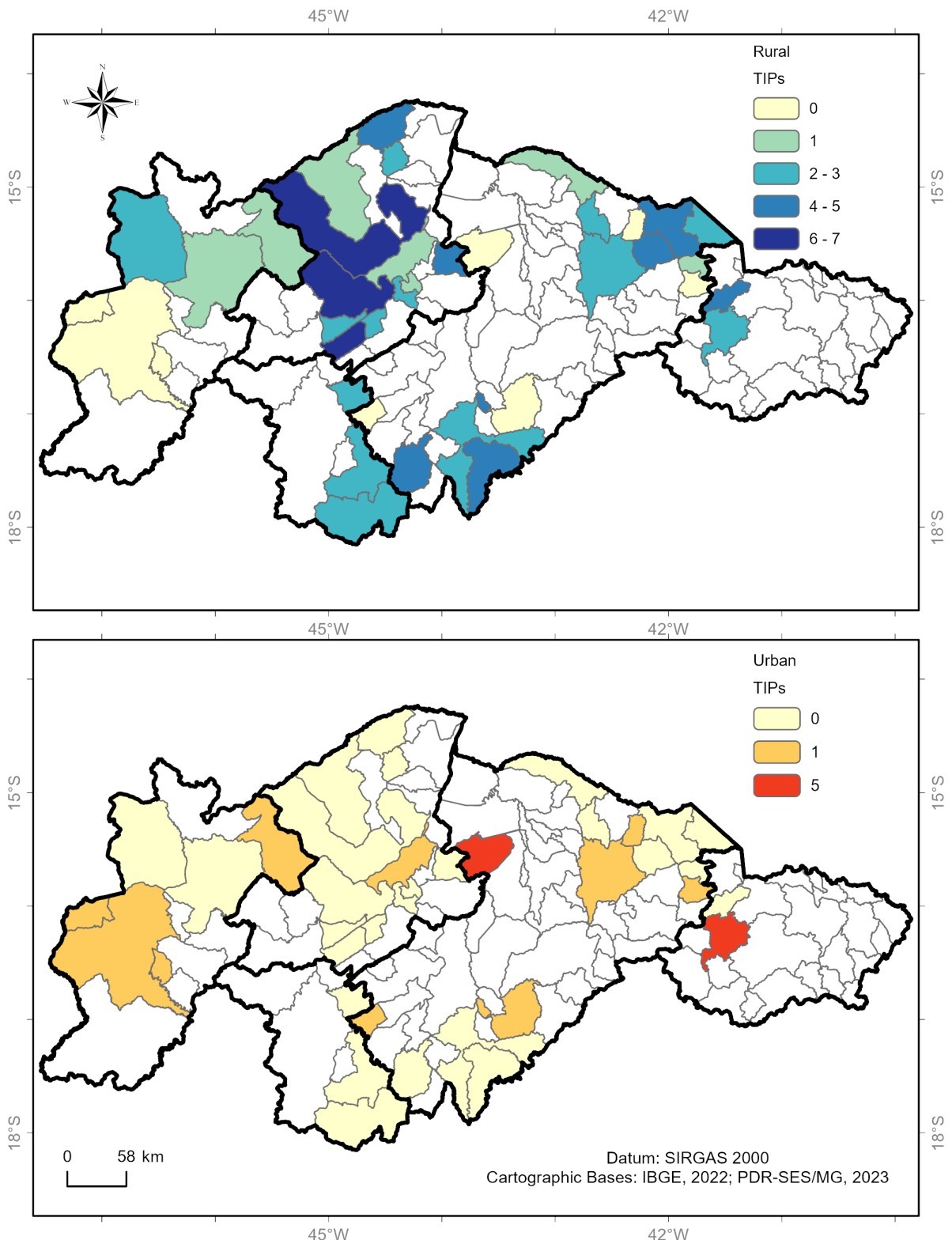

**Fig 13. Spatial distribution of TIPs installed in Family Health Strategy in high-risk areas based on location zone.** Panel A shows the TIPs installed in rural areas, and Panel B shows the number of TIPs installed in urban areas.

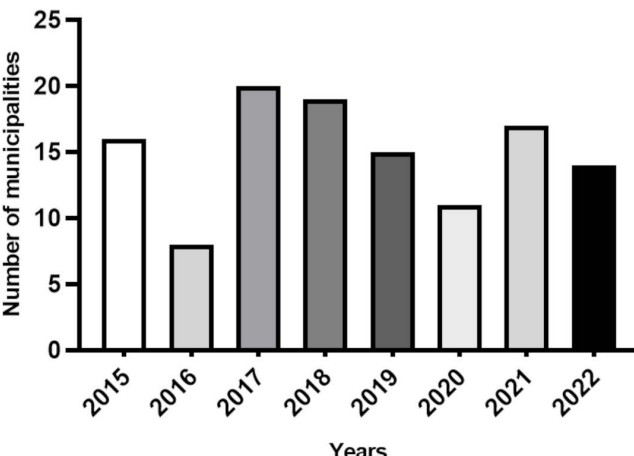

**Fig 14. Number of municipalities reporting the delivery of suspected insects at TIPs between 2015 and 2021 in the Montes Claros RHS.**

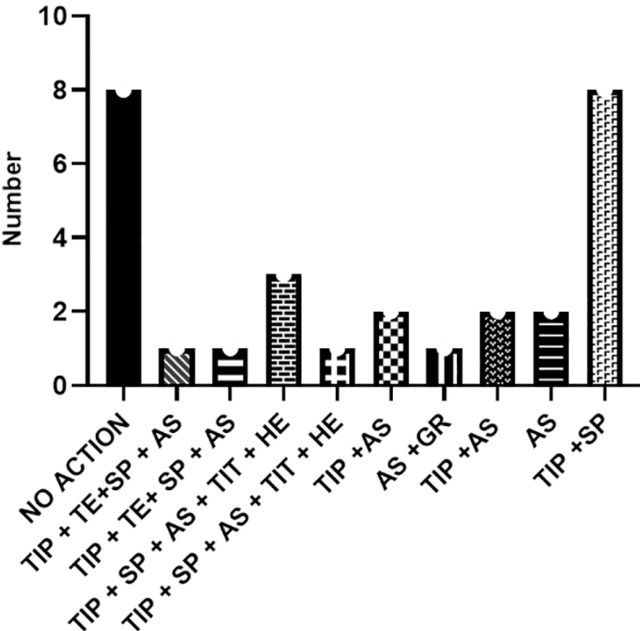

**Fig 15. Number of control actions of Chagas Disease performed in the territories according to the participants.** Codes: TIPs – Triatomine Information Posts; AS – active search; SP – spraying; TE – triatomine examination; GR – geographic reconnaissance; TIT – taxonomic identification of triatomines; HE – health and education.

*"[...] So, a lot of notifications come up, right? But we only have 2 TIPs in **xxxx** and 1 in **xxxx**, so, um, I'm going to, um, open another one, like you were saying, we'll open another one in… **xxxx**, because those people are farther from **xxxx**, right? There's a merchant there, I'm going to talk to him, and he'll join as an independent contractor to help us with the active search for these, um, these triatomines, you know? [...]"*

(FG 1/ Participant 1.3. The proper name was replaced with xxxx.)

*"[...] It was installed, um, it was, um... installed only in the system **(in 2016)**. Because, in that place, the person who works there didn't want to be responsible for receiving the barber bugs, right, and informing us about them. So, um... we pass the information to the health agents, so when people approach them in the rural area regarding this issue, they bring it directly to the city [...] But, for now, um, since 2019, we've been stuck with everything. Chagas, routine, and TIP."* (FG 5/ Participant 5.5) *Words in bold were added by us to the participant's statement*

*"[...] Um... I'm planning to activate one more TIP in a district here in the municipality, which is 22 km away. And then it will be easier for the other distant communities to deliver their triatomine bugs there, to this TIP. So, I'll see if I can reactivate this TIP. Reactivate this TIP in xxxx, which is in this more distant district so that we can resume the work there. But one of the difficulties is the issue of staffing to carry out the work [...] Here in **xxxx**, I do see a need, yes. At least for this TIP in the district, the more distant one I mentioned. In this region, in addition to being a high-infestation area, there are communities 112 kilometers away from the municipal seat. So it's a very far area for people to bring the bugs. So, at least in this district, I do see the need, yes, and that's what we're thinking about reactivating. [...]"* (FG 5/ Participant 5.8. Proper name was replaced with xxxx)

Some municipal endemic disease coordinators highlighted the need to close TIPs, particularly in the rural areas of their municipalities. In this regard, several factors hindering their permanence and sustainability were pointed out, such as lack of knowledge and use of the posts by the population, poor geographical distribution of the posts, abandonment of the TIP by the volunteer collaborators, and lack of feedback on insect examination results to residents. The following excerpts can exemplify this:

*"[...] Yes, I believe so. Because whenever someone comes to me, I tell them: 'No, but there is a TIP in such-and-such area of yours.' Then they would say they didn't know about it. So I believe the information wasn't passed on to them, you know? Maybe the TIP was installed, but the population wasn't informed.[...]"* (FG 5/ Participant 5.10)

*"[...] And also, picking up on what my colleague said, the community here basically... The local communities here became basically disillusioned with the work of the TIPs. So, they go to the headquarters **(epidemiological surveillance)**. Through the headquarters, we also immediately respond about it... We already scheduled a date for the spraying. And, based on that, the work is directed to that, either to a household unit or even to the entire area. Although we are doing active search work, yes, in 100% of the areas of the households. So, um, in any case, is the TIP work important? Yes. But for us here, for our reality, it's only really working in the urban areas. The ones in rural communities, none of them are working, and they haven't been for a long time. Also, even, in my opinion, the population has LOST FAITH in the work of the TIPs, um, of what would be the TIPs in the rural communities here in our municipality.// Yes, in my opinion, from what I've heard from some people, for example, community X, with a TIP installed, the resident would take the triatomine to the TIP, but the person responsible for the TIP, the collaborator, wasn't there, sometimes they were traveling. And so, the lack of commitment to the community. Because it's a voluntary service, and unfortunately, we know that not everything works as agreed upon with people, right? Someone says, 'I commit to being responsible for the TIP here.' But then the resident would arrive with the insect at the TIP and wouldn't find anyone. So, they would kill it there, throw it outside, and that's how it goes. As I said before, well, in my view, we've lost the essence of the main focus of the TIP. Here in our region, I'm talking about **xxxx**. And... Today, here, the headquarters has become the main TIP, and it's where the population really finds the response they are hoping for. Yes, the analysis, sometimes the first analysis of the insect, was even done with the resident present. The person came, like, we have someone here who participated, who did the analysis and already gave the first response right away. So it's much easier, much more convenient to bring it here and get a response than to leave it in the community and then wait two, three, or four months without a plausible response. And when the employee would go to the TIP when they made a visit, they wouldn't find the insect; most of the time, it was already dead, very old, and dry. So, yes, the urban TIP today gave a more effective*

*response in this regard for us.”* (FG5/ Participant 5.1. Proper name was replaced with xxxx). Words in bold were added by us to the participant's statement.

From the participants' perspective, other factors that would justify the deactivation of TIPs include the absence of triatomine occurrences in the localities and the difficulty of access to the TIP for the population, as detailed below:

*“[...] We are considering the possibility of closing some (**TIPs**), right? Because these are areas that have been demolished. We decided to focus on the larger municipality—it's about 3,200 km², as I mentioned. We know there are regions where triatomines are not found, especially in areas that aren't very mountainous [...] Usually, TIPs in those regions tend to yield negative results, not because people don't bring samples, but because historically, there hasn't been much evidence of triatomines. Even so, they remain about 90 km² away from the main hub, making them difficult to access. So, it's important to keep a reference point there. However, we are likely to deactivate a locality that has already been demolished. It was purchased by a major company—a large agribusiness—and the area was cleared. For instance, in xxx, the houses were demolished, and there's no need for a TIP in a place where no one lives, only a firm operates. What we can do is designate that firm as a reference point for bringing samples to us [...]”* (FG 3/ Participant 3.2). Words in bold were added by us to the participant's statement.

Lack of investment in equipment, supplies, and transportation for professionals; unqualified personnel to perform surveillance with community participation; decentralization of endemic control actions; lack of publicity about the posts; and public distrust due to the lack of feedback and effectiveness of the actions implemented were identified as factors hindering the sustainability and maintenance of TIPs in operation in the municipalities covered by the study. The following statements illustrate these issues:

*“[...] Yeah, there were many factors, right? One was the lack of publicity. People started bringing the barber bugs directly to the headquarters, and some of the TIPs were abandoned. From what I've seen, in some places where TIPs were set up, residents have moved, passed away, or no longer live there, you know? So, there were various reasons. But I think the most important one was the lack of publicity because we received a lot of cases here in **xxxx**, and we didn't [...]”* (FG 4/ Participant 4.7. Proper name was replaced with xxxx).

*“[...] Sometimes, before—not recently, but a long time ago—we used to collect the insects, report them, and send them in, but it took a long time to get any feedback on what the results were. Residents would ask a lot of questions, and when we told them, they would say, 'Oh, but it takes too long. What if I'm infected, then what?/... But if the three months pass after I'm infected, how will I be cured? I'll be condemned, just taking medication to control it until it's my turn to die.' So, if there were more investment and better assistance, I think it would be much better. Even the population would have more trust in the TIPs and in the Chagas work carried out by the municipality. [...]”* (FG2/ Participant 2.1)

*“[...] So, what happens? The funding goes to the municipality, but it ends up in the hands of the manager. And, as I mentioned before, sometimes they lack either the information or the awareness of the importance of the work, and they end up diverting investments away from the department. A vehicle, a motorcycle, a staff member, personal protective equipment—it turns into a domino effect. Without the proper structure and necessary investment, you can't deliver quality work. If you can't serve the population, they themselves lose interest in the work because they already know the kind of response they'll get…/ [...] As a colleague mentioned earlier: in the past, there was accountability, right? There was oversight of the work being done. Today, that accountability doesn't exist as much. So, what does the municipality do? They're not obligated to take action. If they don't act, the work doesn't get done, and the population goes unserved. If municipalities were held accountable, if there were stricter enforcement—let's say, when you tie it to funding—the municipality would respond. Funding often comes in, but if the municipality doesn't act, they still receive it, so there's*

*no incentive to take action. What happens then? If you say, 'The municipality will receive X, but only if the work is done,' they would do the work. But if they're going to receive it without doing anything, why would they bother? So, I think investment is fundamental, but municipalities need to be held accountable and required to report their results. If they're held accountable, they'll have to act. It's like in the past when we had those meetings where staff would come to check if targets had been met or not. That doesn't happen anymore. Previously, they monitored Chagas work, dengue work, schistosomiasis work—today, there's no oversight. If municipalities were once again held accountable to ensure the work was carried out and tied funding to performance, I think they would do the work. They'd be monitored and required to report results."* (FG 4/ Participant 4.9)

*"[...] Since the Chagas-related work has stopped, and we're no longer sending anything to xxxx, and xxxx isn't analyzing barber bugs anymore, the population keeps asking, 'What happened with the barber bug?' or 'What was the result with the barber bug?' And we don't have an answer. So, if the work were decentralized, right, and there was proper training for staff—if our entire team were well-trained and our lab technician was qualified—then we'd have better integration with the population. We'd be able to provide better service to them, and the overall health of our municipality would improve. We'd serve everyone better, right?/... I think the best solution is decentralization. Decentralizing the service is, in my view, the best approach. [...]"* (FG 2/ Participant 2.10. Proper name was replaced with xxxx)

Expansion of outreach, mobilization, and engagement of different levels of healthcare and the population, along with the establishment of a regular service flow, were key points highlighted by the focus group participants regarding the TIPs.

*"[...} Triatomine bugs are there, they exist; what's missing is this mobilization, this outreach so that people deliver them, regardless of whether they're positive or not, to either the health unit or the TIP. "With the development of actions, we should include Chagas Disease in the SES mobilization schedule. We have references at the State Health Department, and at RHO as well. So, there should be at least one social mobilization action per bimonthly period regarding Chagas Disease. This includes performing tests, continuing triatomine bug tracking, and moving forward with these actions. And we, here in the municipality, must also do our part, not stop just because of a lack of oversight, but we must also recognize that today, decentralization is part of our commitment to carry out actions and, well, actually implement them, sit down and get things done. I want to thank you all for the opportunity, thank you very much!" [...]* (FG 3/Participant 3.2)

The coordinators identified the following aspects as facilitators for the sustainability and maintenance of active TIPs in the territories: placing the posts in strategic locations such as schools and health units, mobilizing the population and healthcare professionals, and promoting the control actions for Chagas disease (health and education actions). Other factors highlighted as promoters of success in the sustainability of TIPs included the proximity of posts to rural areas and regions with higher triatomine infestation, assistance and visits to TIPs by endemic disease control agents, and investment in training and capacity building for healthcare professionals and TIP volunteer collaborators. The coordinators highlighted the strategic placement of TIPs in schools and health posts as key factors for success in their sustainability. One coordinator explained:

*"[...] In schools, a TIP would be easier for us to... residents to bring the barber bug, so, since we have 10 here, almost all in rural schools, we placed the TIPs in the schools, put up a big sign in front of the school like: 'Here is a point of... a TIP'. Then... notification of TIP. We thought it was better because people, when... had difficulty bringing the barber bug to the city of **xxxx**. So [...] We thought it was better in schools because the kids knew what the barber bug was. When they found the barber bug in the nests or when they went to collect eggs, or found it inside the house, they would give it to their parents. The parents would run to the nearest school, do the registration, and then someone, either the principal or the teacher, would tell us there were six cases of barber bugs at the school, and we would go to the location and*

spray the area. This is how the number of barber bugs in the region started to decrease, you know? Because, you see, we would pass by once a year, the municipality of **xxxx** has 93 locations, so to cover all of them, it would take us two years, with 50% in the first year. When we left a location far away and went to attend the TIPs, we continued like this, expanding the TIPs. Then we found out that health posts were also a good point of reference for people to bring their barber bugs, especially the dog vaccination posts, which worked really well, so we put a TIP there, too. And that... that location.../ [...] So, like I'm saying, the strategic point, in schools, in health posts, a farmer who is friends with everyone, doesn't need to have relatives in the area, as long as everyone likes that person, who is well-received in their home, the TIP would be better at that location." (FG 1/ Participant 1.1. Proper name was replaced with xxxx.)

"[...] The success of the TIP, what really means a lot, is the mobilization of the people who are part of that TIP, right? It's not forgetting that the triatomine exists, it's here, this is its place, and at some point, it will appear. When it does, we have to examine it, right? It might not be a positive case, but if by chance it is, if it's your father, your child, then the mobilization, for me, is the key factor, followed by the visits. If the municipality goes there every month and talks to them, spends an hour, an hour and a half talking to them, they know someone is there to support them. If the visits stop happening, they also won't commit to collecting any barber bugs or making notes of what they find. So, the second important factor for the success of a productive TIP is this permanent visit from the agent, and the third is this periodic meeting between the collaborators. They need to be encouraged, right? The conferences need to remember that the Chagas program exists, in all health conferences on the annual health agenda, they can't forget to plan the visits, to plan the mobilization. Also, this visit **xxxx**, putting it on Instagram, which is the most modern social media tool, is a great partner, but we can't forget, not for a second, about the radio, right? Because people in rural areas don't have much access to Instagram, but they like the radio. So, this issue of putting it on the radio, this whole set of factors to show [...]" (FG 3/ Participant 3.2. Proper name was replaced with xxxx.)

The coordinator highlighted the importance of maintaining vigilance and reinforcing community awareness about Chagas Disease:

"[...] People have this knowledge regarding bringing in the triatomine because I believe that information is very accessible to many; even those living in rural areas today have access to information. Often, people come here almost describing everything, right, about the triatomine, and they are also concerned. So, this issue of permanence, I believe, is more about the work that has already been done previously; it just needs to be reinforced, right, with our presence as surveillance, right, following up on the needs of rural areas and providing support and health education as well in these areas so we can strengthen this work, which is somewhat dormant, right? It's very quiet; it needs to be strengthened in reality. And so, I intend, right, in these interactions we're having, in these conversations, that all of this can be put into practice, right, because I believe it is the desire of many who have worked in the field, who have seen the results of the work, right, for us to be able to return, right, not forgetting our roots while also having this support, this strengthening in the area of Chagas, right, in the issue of Chagas Disease." (FG 3/Participant 3.3)

We identified the professionals' concern about placing the posts in strategic locations that are well-known to the population, particularly in rural areas or near the municipal health surveillance headquarters, making it easier for people to reach the posts. The need to assess the extent of the municipality for installation and the actual demand for the collection and delivery of triatomines was also highlighted. Below is the excerpt:

"[...] We thought z it would be a good idea to place the posts in schools because... each year, there was an evaluation of the students, and some of them knew what barber bugs were. So, the kids gave a talk about barber bugs, and we did creative activities, like going to schools to give talks to parents at the parent-teacher meetings. That's when we

**Table 3. Summary of factors identified by the endemics coordinators as barriers and facilitators to the permanence and sustainability of TIPs in the territories.**

| Hindering factors | Facilitating factors |
| --- | --- |
| Lack of knowledge and use of TIPs by both the population and healthcare professionals. | |
| Need to increase the promotion of TIPs. | Effective promotion of TIPs (engaging the population and healthcare professionals). |
| Need for mobilization and engagement across different health sectors and the population. | Carrying out health and education activities in the territories. |
| Lack of adequately trained professionals to conduct surveillance with community participation. | Investment in training and capacity-building for healthcare professionals and volunteer collaborators working on TIPs. |
| Absence of triatomine occurrences in certain areas. | Prioritization of TIP installations in areas with triatomine occurrences. |
| Abandonment of TIPs by volunteer collaborators. | |
| Placement of TIPs in hard-to-reach areas. | Strategic placement of TIPs in key locations (e.g., rural areas of territories). |
| Lack of feedback on insect examination results to residents. | Support and visits to TIPs by endemic disease control agents in certain territories. |
| Distrust of the population regarding health surveillance services. | |
| Lack of investment in equipment, supplies, and transportation for professionals. | |
| Decentralization of endemic disease control actions. | |

*realized that if we placed the posts in certain farm areas, where the residents were friends with everyone, that was the best choice. For example, if someone threw a 'folia de reis'* **(a traditional celebration)***, everyone would go there, and that person was well-liked. We decided to place the TIP there. It didn't make sense to place a TIP in a rich person's house if they weren't friendly with the neighbors, because no one would go to that house. So, we chose people from the region who were humble and liked by everyone, and the neighbors would bring the barber bugs to them. That's when we thought it would work better this way, you know? Because we first chose people from the region, people who were well-liked by the community."* (FG 1/ Participant 1.1) *Words in bold were added by us to the participant's statement.*

Table 3 summarizes all the factors identified by the endemics coordinators as barriers and facilitators to the permanence and sustainability of TIPs in the territories. These aspects complement each other.

## Discussion

The current challenge in Chagas disease in Brazil, and likely in other endemic areas of Latin America, aiming for long-term control of domestic reinfestation by native vectors, can only be achieved through a balance of technical and political-administrative aspects in a sustainable epidemiological surveillance system. This system should include horizontal strategies in the decentralization model of actions, as well as constant supervision and broad community participation [8,28].

Dias in 2000 [25] stated that the main actors in the vector surveillance of Chagas disease in Brazil are local health and education systems, public health institutions, and the community itself. In this context, working in one of the regions with the highest vulnerability to Chagas disease in the world, the present study focused on understanding the structure of the primary tool for surveillance with community participation: the TIPs. Today, in Brazil, these posts support and anchor surveillance with community participation, with vector surveillance in the field being almost entirely restricted to them [25]. However, according to the same author, in the year of the referenced study, TIPs were still subject to low or no supervision by government entities [25]. Over 20 years after the statement made by the author, this study found that even in areas where the strong operation of surveillance with community participation is crucial, the information found about the TIPs is fragmented and inconsistent.

We observed that most municipalities in the study area did not maintain documented records of TIP usage and productivity; when they did, it was in a very rudimentary form, revealing weaknesses in the SISPCDCh, which provides limited and basic information. Additionally, a significant portion of the municipalities in the study area had never had a TIP installed in their territory. Even among those that had installed TIPs, a considerable portion was deactivated, particularly in rural areas of the municipalities. In this context, the endemics coordinators from the municipalities in the study area highlighted a range of facilitating and hindering factors for the functioning and sustainability of these posts in the territories, which are discussed below.

## Physical, situational, and spatial characterization of TIPs

At the outset of the research, several concerning issues were identified concerning aspects related to the management, documentation, and monitoring of activities and actions for surveillance with community participation in the study area. Accurate data collection was hindered, as 100% of the municipalities did not use any of the forms and records recommended by the Brazilian Ministry of Health [11,12] for monitoring and operating TIPs. Prevention strategies and control and monitoring programs for specific diseases and vectors require reliable information about the situation of these diseases and the distribution and notification of vectors in the territories. In the absence or inadequacy of information on collection, processing, notification, analysis, and data interpretation, it becomes difficult to devise effective actions for disease prevention, control, and monitoring, as well as for vector management [29].

The number of municipalities in the study area that had never had TIPs installed was notably high, especially in the health regions of Pedra Azul and Montes Claros, considering primarily the fact that these areas are classified as high-risk zones for the domestic reinfestation of triatomine bugs. According to a previous exploratory study conducted in the same area and year as the current research, the authors found that most municipalities had significant and various gaps related to the management of Chagas disease control activities, including complete halts in actions [30]. The Montes Claros RHS and Januária RHO stood out in this regard, with over 96% of their municipalities experiencing activity suspensions [30]. Additionally, only around 20% of the municipalities in the Pedra Azul, Pirapora, and Januária RHOs were conducting at least one control action [28]. Finally, only 28% and 35% of the municipalities in the Pedra Azul RHO and Montes Claros RHS had endemic disease control agents involved in Chagas disease control actions, respectively. There were endemics coordinators for the disease in 24% and 48% of the municipalities in the Pedra Azul and Januária RHOs, respectively [30].

Given the incipient management of control and surveillance actions observed in a large part of the municipalities in the study area, coupled with the scarcity/reduction of the local technical staff, the strengthening of surveillance with community participation becomes an even more relevant strategy, considering the biodiversity and the presence of triatomines in the local scenario.

In general, the number of TIPs installed in the study area, as well as their concentration by municipality, reflects the effort of local surveillance services to concentrate these posts in areas of higher risk for Chagas disease, although many municipalities were left uncovered. The vast majority of TIPs were installed in rural areas of the municipalities, as expected since it is in these areas that residents are more exposed to contact with triatomines. Despite the expansion of the epidemiological scenario of Chagas disease in recent years in Brazil, residents of endemic rural areas remain the most vulnerable segment of the population to Chagas infection [31,32].

The heterogeneity in the installation and concentration of TIPs likely originates from modifications in the form and organization of the teams' work after the decentralization process of healthcare [33]. Previously, vector control programs had a vertical structure, directly executed by agencies linked to the Brazilian Ministry of Health, so actions occurred more uniformly across territories. With the transfer of programs and responsibilities (operation, planning, and execution of health activities) to states and municipalities [33], actions began to occur in a more fragmented and dispersed manner across territories, as already discussed in previous studies [32].

The analysis of the operational situation of TIPs revealed a significant problem in maintaining these posts, as one-third of them were deactivated at the time of data collection. This aspect proved particularly important in the Januária RHO, which had nearly 60% of its TIPs deactivated, followed by the Pedra Azul RHO. In comparison, other health regions showed higher rates of active TIPs, reflecting better capacity for maintenance and sustainability of these posts by local services. An aggravating factor to the scenario of deactivated TIPs in the region is the fact that the vast majority of these deactivations occurred in posts located in rural areas (81.7%), which is even higher than what would be expected by chance.

As mentioned by the focus group participants, rural areas face specific challenges to the sustainability of TIPs, such as frequent population migration and the consequent depopulation of localities; lack of adequate resources for the transportation of health professionals; and logistical difficulties in remote areas, which compromise the necessary support and monitoring for the maintenance of TIPs in operation. The deactivation of TIPs in rural areas is particularly concerning, as it undermines the surveillance and control of Chagas disease in the most vulnerable regions. In this regard, it is essential to develop specific strategies that address the particularities of rural areas and ensure the sustainability of TIPs, in addition to strengthening the infrastructure and technical support necessary to operate these posts in high-risk areas.

The installation of TIPs in municipalities from almost all health regions occurred primarily in public institutions (such as schools and health surveillance centers, among others), followed by residential installations. In contrast, in the Pedra Azul RHO, there was a higher number of TIP installations in residences than in public institutions. The installation of TIPs in commercial establishments was low throughout the study area. These findings align with the statements by Moreno [9], who indicated that the preferred locations for TIP installation are schools and health posts, and in the absence of these, in the residences of community leaders and reference individuals.

Among the TIPs installed in public spaces, those located in common areas of the Family Health Strategy stood out, particularly because installing the posts in these spaces seems to be a key factor in keeping them active, a hypothesis that was statistically supported. These findings were even more evident in the Januária RHO and Montes Claros RHS, where high rates of TIPs located in active Family Health Strategy were observed compared to TIPs installed in areas outside the Family Health Strategy. In this regard, no differences were detected in the Unaí and Pirapora RHOs. In contrast, in the Pedra Azul RHO, 100% of the TIPs located within the Family Health Strategy were deactivated.

Our results highlight that the location of TIPs in common areas of Family Health Strategy influences in the permanence and sustainability of the posts. Dias [25] emphasized that entities such as family health programs, community health agents, and inter-municipal consortia are facilitating elements in enabling epidemiological surveillance of Chagas disease within the SUS. In this context, it is believed that the presence of community health agents, professionals who bridge Primary Healthcare Units and the population [34], plays a significant role in strengthening health promotion actions, particularly in the context of health surveillance for Chagas disease, as previously demonstrated in a study that discusses the role of community health agents and their reception by the population [19]. In summary, it is suggested that the effective integration of TIPs with physical spaces shared with Family Health Strategy strengthens the epidemiological surveillance network and enhances the response to disease control, especially in regions where PHC engagement is more effective within territories [35].

### Characterization of the productivity and use of TIPs

The lack of documentation regarding the productivity of TIPs observed in most municipalities worsens the critical scenario of community-based surveillance in the study area, considering that triatomine infestation rates in households are the primary indicator used in decision-making [36] by governmental and health authorities. Therefore, the mere existence of TIPs in the territories is not enough; the population must be engaged in the search for and use of these posts.

The absence of clear and standardized records on the activity of the posts in all health regions, except for the Montes Claros RHO, creates a chaotic scenario, making it impossible to monitor and, consequently, evaluate the effectiveness of

Diseases

community-based control actions for Chagas disease. It is important to highlight that vector surveillance, here viewed as community-based surveillance, primarily aims to detect and eliminate household infestation foci, allowing for the monitoring of reinfestation in areas considered under control [8; 37–39]. However, an efficient documentation system is essential to enable data tracking and recovery to achieve ideal monitoring and subsequent detection/elimination of foci.

Although the Montes Claros health region was recording TIP activities in the information system, the number of installed and active posts in the area (163) did not translate into high productivity. During the assessed period, the registration and notification of suspected insect receptions were low and disproportionate to the installed potential, especially considering that the region is located in an area of high risk for reinfestation of triatomines. The contrast between the existing infrastructure and the low productivity suggests failures in management and local team involvement, aspects already described in the literature for the study area [30], which may be exacerbated by the lack of an adequate system for registering and controlling triatomine notifications.

This discrepancy highlights the urgent need to strengthen not only the maintenance and operation of TIPs but also to implement an efficient documentation and information system, ensuring that productivity aligns with the installed capacity in the territories, especially in high-vulnerability regions. In this regard, Moreno [9] discussed the importance of maintaining a vector information system that manages and ensures periodic fieldwork monitoring, allowing for the tracking and evaluation of results and facilitating changes and adjustments in control and surveillance strategies.

## Perception of health surveillance professionals about the functioning of TIPs in the territories

The first highly relevant aspect gathered from the statements of several participants was the perception of a lack of knowledge regarding the functioning and/or existence of TIPs among both surveillance professionals and residents. Undoubtedly, this is the main cause of the non-use of TIPs in the territories, explaining the high number of TIPs that were deactivated in the municipalities. Furthermore, those still in operation have low productivity, as shown by the productivity data for TIPs located in the Montes Claros RHS. In response to this, the participants themselves mentioned that a factor that would contribute to the sustainability of TIPs in the territories would be the implementation of health education actions, as well as proper dissemination of the posts, mobilization, and engagement of healthcare professionals and residents. In an extensive literature review on community participation in Chagas disease surveillance, Abad-Franch et al. [8] highlighted that most surveillance experiences conducted in various Latin American countries did not involve the community in designing, planning, and evaluating interventions. The authors argued that the genuine involvement of all stakeholders throughout the process would promote the effective empowerment of the community. In the present study, something similar to what the authors observed and discussed is shown and suggested, supporting what was mentioned by professionals regarding the need for mobilization and engagement of the different sectors of healthcare and the population in the process of community-based surveillance and the use of TIPs.

The scarcity of health education actions involving the vector surveillance of Chagas disease is a reality already shown in other areas of Brazil [32]. However, surveillance based on the community, enriching collective discussions, and promoting knowledge through educational actions that engage the entire community are essential for individuals to exercise their autonomy and truly become empowered [40].

Beyond the mobilization and engagement of the population, another aspect that emerges as a concern for the sustainability of surveillance and TIPs is maintaining the motivation and interest of the population to continue participating in surveillance in areas with a progressive decrease in reported cases of Chagas disease and domiciled triatomines [41,42]. In the present study, several participants pointed out that a major barrier to the use of TIPs in the territories, and a consequent trigger for the deactivation of these posts, is the absence of triatomines in the areas. Abad-Franch et al. (2011) [8] indicated that sustaining community awareness becomes a significant challenge in increasingly rare infestation events. In this regard, the authors suggested continuous education for health professionals, and once again, these insights align with the present findings. During the analysis of the focus groups, endemic disease coordinators highlighted the lack of

properly qualified professionals to engage in surveillance with community participation, emphasizing the need for investment in training and capacity building for health professionals and the volunteer collaborators within the TIPs.

Bringing to light the volunteer collaborators within the TIPs, there were statements mentioning the abandonment of the posts by these collaborators prior to the COVID-19 pandemic, leading to their inactivity. Silva et al. (2022) [24] administered a questionnaire to the TIP collaborators in the municipalities studied here and found that 49% had not received any instructions or training to perform this role and activity. They suggested that one of the motivators for the abandonment of the posts by the collaborators is the lack of training for the role, combined with other factors such as changes in the living conditions of the Brazilian population, which hinder the permanence of individuals in these positions as volunteers, as well as changes in the spatial and social relations brought about by the use of new technologies, including internet and social media. In this regard, we suggest that one of the key changes required for the surveillance system with community participation, structured through TIPs in Brazil, will be the reconfiguration of the role and responsibilities of the volunteer collaborators within the TIPs.

Another factor highlighted by participants and identified as a challenge to the use of TIPs was their poor geographical distribution across the territories. It was mentioned that installing TIPs in strategic locations, considering factors such as the size of the municipality, rural areas, proximity to the municipal seat, and schools, among others, would facilitate their permanence and use. In contrast, some participants noted that the installation of TIPs in rural areas is a hindrance and should be avoided. In this regard, the need to expand the network of TIPs was also mentioned, considering that some municipalities have large territorial extensions, and the distance between one TIP and another poses a challenge.

According to previously collected data, as discussed above, the TIPs in the studied municipalities were correctly installed, following the criteria defined by the Brazilian Ministry of Health in the Superintendence of Public Health Campaigns (*Superintendência de Campanhas de Saúde Pública* – SUCAM) manual for TIP installation [11,12]. They were installed in urban areas and rural localities of municipalities with significant population, epidemiological, and geographical relevance. Additionally, they were preferably installed in collaboration with community leaders (in residences or businesses), schools, near health surveillance sectors, and Primary Healthcare Units. Moreover, in the studied area, and as previously discussed, the strategic importance of installing TIPs in spaces shared with the Family Health Strategy was observed. Supporting these findings, Dias (2000) [25] emphasized that the vector surveillance of Chagas disease should be sustained and expanded at different levels of health promotion.

A very delicate and concerning issue highlighted by participants was the lack of feedback to residents regarding the insect test results. In this regard, it was mentioned that the visit of endemic disease control agents to the TIPs would facilitate the sustainability of the TIPs. This is suggested to be one of the most serious problems to be addressed by community-based surveillance in Brazil, as the lack of a prompt response and also the lack of vector control activities disempowers and disengages the community, even leading to a loss of trust in health surveillance services, another hindrance noted by the coordinators. Regarding this matter, Dias (2000) [25] stated that key elements of vector surveillance include the logistics and strategy of the system, as well as the speed of response to notifications and emerging problems. The author clarified that the roles of all involved actors should be clearly defined from the start and throughout the surveillance process, involving technical elements and leadership. Likewise, all notifications and issues should receive a response, which is essential to sustaining community participation.

Abad-Franch et al. (2011) [8] further emphasized that there must be a defined communication channel between residents and endemic disease control agents for the long-term success of community-based surveillance, enabling a timely response to any notifications, even for non-triatomine insects [43,44]. In summary, control programs should therefore incorporate community-based approaches from the outset, including a professional, prompt, and timely response to each notification, thereby reinforcing and maintaining the community empowerment process.

Dias & Garcia (1978) [45] showed that the faster the intervention in triatomine foci, with appropriate epidemiological measures, the greater the resulting benefit, reinforcing system coherence and community participation. However, for this

harmony to be achieved, surveillance professionals and the service itself must be technically and structurally prepared to provide these prompt responses. This does not seem to be the case in the studied territories, where there were reports regarding the lack of investment in equipment, supplies, and transportation for the professionals.

Finally, some statements brought up the issue of the decentralization of health actions that took place in Brazil in 1999 [33], which is seen as a limiting factor for the full functioning of community-based surveillance and TIPs in the territories. This is an aspect already discussed in the scientific literature in other contexts vector surveillance. Souza et al. (2023) [46] conducted a focus group with endemic disease coordinators in municipalities in Minas Gerais and reported that participants identified the decentralization of health as a trigger for profound changes in how vector surveillance teams for Chagas disease were organized and operated. In contrast, it is important to emphasize that municipal and regional actions are technically feasible [25]. However, they depend on good organization, competence, and sustainability mechanisms. In this context, the lack of resources and the low prioritization of control measures by local managers undermine the effectiveness of control actions, resulting in failures in the control and surveillance of Chagas disease.

In 2021, a comprehensive diagnosis was conducted involving the technical references at the state level in Brazil responsible for the vector surveillance of Chagas disease. Through this diagnosis, a document [47] was developed, outlining actions and recommendations, including the promotion of local leadership in the planning and prioritization of vector surveillance actions for Chagas disease. This approach aims to strengthen regionalization and decentralization of actions, in line with Consolidation Ordinance 4/2017 and complementary regulations [11; 48–52].

## Final considerations

The present study was conducted during the COVID-19 pandemic, which may have seemingly influenced the responses and the situation detected in the municipalities. However, during the focus groups, it became evident that the issues highlighted here have been longstanding and chronic in the health services. Moreover, most TIPs had been deactivated and unproductive for many years in the territories.

We revealed a scenario of heterogeneous distribution of TIPs across the territories, with some areas having a higher number of TIPs installed compared to others. However, this did not necessarily reflect the productivity of these posts, as many were found to be deactivated, and even those in operation showed low productivity. We suggest that the association of TIPs with the Family Health Strategy services could be a strategy to ensure the long-term sustainability of the posts, and therefore, the strengthening of this integration should be encouraged throughout Brazil. The lack of documentation and traceability of data was a recurring issue, which hinders monitoring notifications in TIPs and, consequently, the use of data in local decision-making.

Regarding the factors that need to be improved, strengthened, and enhanced, it is important first to highlight that the organization of health systems, education, and local social control committees facilitates the regular functioning of surveillance with community participation. In addition, the population will only engage in surveillance if they receive quick and satisfactory responses from local surveillance services in return for their involvement. For this engagement to occur, the population needs to be sensitized and mobilized to care for their households. In this regard, health education should be presented from a political-pedagogical perspective, contributing to the development of a critical and reflective stance on reality, with people seen as historical and social subjects capable of transforming their lives and the community.

## Supporting information

**S1 Fig. Spatial distribution of TIPs based on the type of installation site.** Panel A shows the number of TIPs installed in commercial locations. Panel B displays the number of TIPs installed in public institutions, and Panel C presents the number of TIPs installed in residences. Municipal headquarters are indicated by black dots, while the centers of the RHUs are represented by white dots with black outlines. Hatched municipalities did not respond to the data collection. (TIF)

**S2 Fig. Spatialization of the TIPs deactivated in the study area based on the geographic zone.** Panel A presents the number of deactivated TIPs in rural areas, while Panel B highlights the number of deactivated TIPs in urban areas.
(TIF)

**S3 Fig. Spatial distribution of TIPs that remained active in the study area according to the geographical zone.** Panel A presents the number of active TIPs in rural areas, while Panel B highlights the number of active TIPs in urban areas.
(TIF)

**S4 Fig. Distribution and number of TIPs in rural areas based on operational status in areas shared with Family Health Strategy across the five health regions.** A, active; B, deactivated.
(TIF)

**S5 Fig. Distribution of TIPs in urban areas based on operational status (active/deactivated) in areas shared with Family Health Strategy across the five health regions.** A, active; B, deactivated.
(TIF)

**S1 Table. Profile of participants in the different focus groups (FG) according to role and sex.**
(DOCX)

**S2 Table. Time in the position of endemics coordination.** FG: focus groups.
(DOCX)

**S3 Table. Number of participants in the different focus groups (FG)who mentioned encountering triatomines in the territories.**
(DOCX)

## Acknowledgments

We would like to thank the the RHOs of Pedra Azul, Pirapora, Unaí, Januária, and the Montes Claros RHS. Our deepest gratitude goes to all the healthcare professionals in the study area who contributed to and enriched our work. Finally, we thank the contributions and suggestions from the reviewers of PPSUS/FAPEMIG/2020: Alberto Novaes Ramos Júnior and Stephanie Gazzinelli.

## Author contributions

**Conceptualization:** Raquel Aparecida Ferreira.

**Data curation:** Valéria Carla Faria Amaral, Millena Vieira Simões de Freitas, Raquel Aparecida Ferreira.

**Formal analysis:** Valéria Carla Faria Amaral, Millena Vieira Simões de Freitas, Gustavo Libério de Paulo, Silvia Ermelinda Barbosa, Bruno Silva Amaral, Raquel Aparecida Ferreira.

**Funding acquisition:** Raquel Aparecida Ferreira.

**Investigation:** Valéria Carla Faria Amaral, Millena Vieira Simões de Freitas, Raquel Aparecida Ferreira.

**Methodology:** Gustavo Libério de Paulo, Raquel Aparecida Ferreira.

**Project administration:** Valéria Carla Faria Amaral, Raquel Aparecida Ferreira.

**Supervision:** Raquel Aparecida Ferreira.

**Validation:** Valéria Carla Faria Amaral, Millena Vieira Simões de Freitas, Gustavo Libério de Paulo, Silvia Ermelinda Barbosa, Janice Maria Borba de Souza, Bruno Silva Amaral, Liléia Gonçalves Diotaiuti, Raquel Aparecida Ferreira.

**Visualization:** Valéria Carla Faria Amaral, Millena Vieira Simões de Freitas, Gustavo Libério de Paulo, Silvia Ermelinda Barbosa, Janice Maria Borba de Souza, Bruno Silva Amaral, Liléia Gonçalves Diotaiuti, Raquel Aparecida Ferreira.

**Writing – original draft:** Valéria Carla Faria Amaral, Millena Vieira Simões de Freitas, Raquel Aparecida Ferreira.

**Writing – review & editing:** Gustavo Libério de Paulo, Silvia Ermelinda Barbosa, Janice Maria Borba de Souza, Bruno Silva Amaral, Liléia Gonçalves Diotaiuti, Raquel Aparecida Ferreira.

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
