## [Decision Letter · Decision Letter 0]

PNTD-D-24-01922Community-based surveillance of Chagas Disease: characterization and use of triatomine information posts (TIPs) in a high-risk area for triatomine reinfestation in Latin America Triatomine information posts in an endemic area for Chagas diseasePLOS Neglected Tropical Diseases Dear Dr. Ferreira, Thank you for submitting your manuscript to PLOS Neglected Tropical Diseases. After careful consideration, we feel that it has merit but does not fully meet PLOS Neglected Tropical Diseases's publication criteria as it currently stands. Therefore, we invite you to submit a revised version of the manuscript that addresses the points raised during the review process. Please submit your revised manuscript within 30 days May 10 2025 11:59PM. If you will need more time than this to complete your revisions, please reply to this message or contact the journal office at plosntds@plos.org. Please include the following items when submitting your revised manuscript: * A rebuttal letter that responds to each point raised by the editor and reviewer(s). You should upload this letter as a separate file labeled 'Response to Reviewers '. This file does not need to include responses to any formatting updates and technical items listed in the 'Journal Requirements' section below. * A marked-up copy of your manuscript that highlights changes made to the original version. You should upload this as a separate file labeled 'Revised Manuscript with Track Changes '. * An unmarked version of your revised paper without tracked changes. You should upload this as a separate file labeled 'Manuscript '. If you would like to make changes to your financial disclosure, competing interests statement, or data availability statement, please make these updates within the submission form at the time of resubmission. Guidelines for resubmitting your figure files are available below the reviewer comments at the end of this letter. We look forward to receiving your revised manuscript. Kind regards, Eric Dumonteil, Ph.D.Academic EditorPLOS Neglected Tropical Diseases

Audrey Lenhart

Section Editor

Shaden Kamhawi

co-Editor-in-Chief

Paul Brindley

co-Editor-in-Chief

**Journal Requirements:**

At this stage, the following Authors/Authors require contributions: Valéria Carla Faria Amaral, Millena Vieira Simões de Freitas, Gustavo Libério de Paulo, Silvia Ermelinda Barbosa, Janice Maria Borba de Souza, Bruno Silva Amaral, Liléia Gonçalves Diotaiuti, and Raquel Ferreira. Please ensure that the full contributions of each author are acknowledged in the "Add/Edit/Remove Authors" section of our submission form.

2) Please ensure that the funders and grant numbers match between the Financial Disclosure field and the Funding Information tab in your submission form. Note that the funders must be provided in the same order in both places as well.

**Reviewers' comments:** Reviewer's Responses to Questions

**Key Review Criteria Required for Acceptance?**

**Methods:**

-Are the objectives of the study clearly articulated with a clear testable hypothesis stated?

-Is the study design appropriate to address the stated objectives?

-Is the population clearly described and appropriate for the hypothesis being tested?

-Is the sample size sufficient to ensure adequate power to address the hypothesis being tested?

-Were correct statistical analysis used to support conclusions?

-Are there concerns about ethical or regulatory requirements being met?

Reviewer #1: The experimental design is correct and the results are based on clear statistical analysis. The discussion has been developed on the basis of the interpretation of the results.The authors have clearly mentioned the ethical endorsement made.

Reviewer #2: This study adopted a mixed-methods approach, with both objectives and approaches applied to health using an exploratory research design, encompassing both descriptive and analytical forms of research and is appropiate to adress the objectives, also de population and study area is appropiate described.

Reviewer #3: The methods would benefit from an overview of the research questions and study objectives and the steps taken to achieve each objective. These could be emphasized more throughout the whole paper and used to structure the paper.

**Results:**

-Does the analysis presented match the analysis plan?

-Are the results clearly and completely presented?

-Are the figures (Tables, Images) of sufficient quality for clarity?

Reviewer #1: The results and analysis have been described correctly. There is an excessive use of figures and it is not clear which will be part of the text and which will be supplementary information.

Reviewer #2: The study mets all the parameters for this section

Reviewer #3: The results would also benefit from more structure. Please organize the results section by the research question/objective that each result answers, using the question or theme as a subheadings.

L504-509 Give a brief summary of each item listed

L526: Put information in table S3 into the main text

In general, much of the information that is stated to be in supplementary material would be better in the main text.

In the perceptions of health surveillance professionals section, the quotes are valuable, but they are too numerous and too long, such that they lose their effectiveness. I suggest reducing the number of quotes and also reducing their length so that the point they are meant to illustrate is clearer. Also, more subheadings would be useful here to help guide the reader through the plethora of content.

**Conclusions:**

-Are the conclusions supported by the data presented?

-Are the limitations of analysis clearly described?

-Do the authors discuss how these data can be helpful to advance our understanding of the topic under study?

-Is public health relevance addressed?

Reviewer #1: In the conclusions the authors detail the contributions of the study and the information and production undertaken to support the reactivation of TIPs, the access constraints created by the pandemic were well explained and the procedures used to address them.

Reviewer #2: The study mets all the parameters for this section. In the PDF I suggest the inclusion of some similar studies in Colombia and Guatemala

Reviewer #3: The conclusions are supported by the data. The discussion section lacks structure and would benefit from more thematic subheadings to guide the reader.

**Editorial and Data Presentation Modifications?**

Reviewer #1: (No Response)

Reviewer #2: This is a good research with well presented methodology and results. I include some comments in the PDF and recommend minor revision

Reviewer #3: (No Response)

**Summary and General Comments:**

Reviewer #1: The study provides a reliable description of the operation, structure and use of TIPs in a highly endemic area of Chagas disease in Brazil since the 1990s. It is an important study because it evaluates, over a significant period of time, the entomological surveillance system that has been established as an effective tool in other countries. The successes and limitations that emerge over time will be of great value to control programmes in the region. The study falls within the scope of the journal, but for publication I suggest some adjustments to the authors.

Introduction: Line 86 mentions passive surveillance but does not describe it properly, I suggest the authors include a brief description of it.

Methodology: Line 179 should be consistent in the use of numbers or letters.

Results:

The number of figures is excessive and it is not clear in the text which figures should be added and which should be included in the text. Authors are advised to keep the more complex figures in the text and the rest as supplementary information.

For example: Figures 6, 7 and 8 could be summarised in a table as well as Figures 10, 11 and 12.

General comments

The experimental design is correct and the results are based on clear statistical analysis. The discussion has been developed on the basis of the interpretation of the results.

Reviewer #2: This is a good research with well presented methodology and results. I include some comments in the PDF and recommend minor revision

Reviewer #3: This is an important evidence-based demonstration of the realities and challenges of Chagas disease surveillance. The information presented will be useful for future researchers. The manuscript text is well-written, but at times hard to follow because there is so much information presented and not enough structure to help the reader keep track of it. The impact of the paper will greatly increase if the authors insert guideposts into the paper in the form of clear objectives and/or research questions and then framing the methods, results, and discussion in terms of those objectives or themes. The paper would greatly benefit from more subheadings as well.

A second critical change that must be made is reduction of acronyms. The paper uses too many acronyms, making it impossible to follow in many paragraphs (VC, CD, PHU, CHA, RHO, RHS, EDCA, SISPCDCh, FHS, TIP, etc.) The number of acronyms used must be reduced to two or three at most.

PLOS authors have the option to publish the peer review history of their article (what does this mean? ). If published, this will include your full peer review and any attached files.

**Do you want your identity to be public for this peer review?** For information about this choice, including consent withdrawal, please see our Privacy Policy .

Reviewer #1: No

Reviewer #2: **Yes: ** Gabriel Parra-Henao

Reviewer #3: No

---

## [Decision Letter · Decision Letter 1]

PNTD-D-24-01922R1

Community-based surveillance of Chagas Disease: characterization and use of triatomine information posts (TIPs) in a high-risk area for triatomine reinfestation in Latin America Triatomine information posts in an endemic area for Chagas disease

Dear Dr. Ferreira,

Thank you for submitting your manuscript to PLOS Neglected Tropical Diseases. After careful consideration, we feel that it has merit but does not fully meet PLOS Neglected Tropical Diseases's publication criteria as it currently stands. Therefore, we invite you to submit a revised version of the manuscript that addresses the points raised during the review process.

Please submit your revised manuscript within 60 days Jun 05 2025 11:59PM. If you will need more time than this to complete your revisions, please reply to this message or contact the journal office at plosntds@plos.org. Please include the following items when submitting your revised manuscript:

We look forward to receiving your revised manuscript.

Kind regards,

Eric Dumonteil, Ph.D.

Academic Editor

Audrey Lenhart

Section Editor

Shaden Kamhawi

co-Editor-in-Chief

Paul Brindley

co-Editor-in-Chief

**Reviewers' Comments:**

Reviewer's Responses to Questions

**Key Review Criteria Required for Acceptance?**

**Methods**

-Are the objectives of the study clearly articulated with a clear testable hypothesis stated?

-Is the study design appropriate to address the stated objectives?

-Is the population clearly described and appropriate for the hypothesis being tested?

-Is the sample size sufficient to ensure adequate power to address the hypothesis being tested?

-Were correct statistical analysis used to support conclusions?

-Are there concerns about ethical or regulatory requirements being met?

Reviewer #1: The experimental design is correct and the results are based on clear statistical analysis. The discussion has been developed on the basis of the interpretation of the results.The authors have clearly mentioned the ethical endorsement made.

Reviewer #4: This work is a case study with both qualitative and quantitative approaches that analyzes the physical, structural, situational, and spatial characteristics of the TIPs (Territories of Health Promotion), the productivity and engagement of the population, the relevance of the Family Health Strategy in health surveillance with popular participation, and the perceptions of health professionals regarding the functioning and sustainability of the TIPs.

The study presents a research design that is appropriate for addressing the proposed objectives. The study population is clearly described, and the sample size appears sufficient to support the intended analysis.

However, the statistical methods proposed for analyzing the data are quite basic. The study does not include any advanced analytical strategies that could model the relationships among key variables such as the conditions of the TIPs, productivity levels, environmental contexts, or professional perceptions. In this sense, the study is limited to simple comparisons to meet its objectives, which restricts the depth of its analytical insights.

This study was approved by the Research Ethics Committee of the René Rachou Institute.

Reviewer #5: Yes, the objectives of the study is clearly articulated with a clear testable hypothesis;

Yes, the study is design appropriate to address the stated objectives;

Yes, the population is clearly described and appropriate for the tested hypothesis;

Yes, the sample size is sufficient to ensure adequate power to address the tested hypothesis;

Yes, The correct statistical analysis was used to support the conclusions;

Yes, all the concerns about ethical or regulatory requirements are being met.

**Results**

-Does the analysis presented match the analysis plan?

-Are the results clearly and completely presented?

-Are the figures (Tables, Images) of sufficient quality for clarity?

Reviewer #1: The results and analysis have been described correctly. Analysis has been developed on the basis of the interpretation of the results.

Reviewer #4: The analysis presented aligns with the general objectives of the study, particularly in describing and comparing key characteristics of the TIPs, community engagement, and professional perceptions. However, while the study adopts both qualitative and quantitative approaches, the quantitative analysis is limited to basic descriptive and comparative methods. No advanced statistical modeling or in-depth analytical strategies are included, despite the potential for richer insights through such methods. Therefore, although the analysis matches the general plan, it does not fully exploit the possibilities offered by the data collected.

The results are presented clearly in terms of basic findings and descriptive patterns. The study adequately describe and reports the physical and structural situation of the TIPs, the location of the TIPs and the characterization of the use and productivity the TIPs. However, the completeness of the results is limited by the simplicity of the analytical methods. Without deeper modeling or exploration of relationships between variables, some potentially important insights—such as factors influencing TIP sustainability or the impact of community engagement—are left underexplored. While the results are clearly presented, the analysis does not fully leverage the available data. The study relies on basic descriptive comparisons, which limits its ability to explore deeper relationships and extract more nuanced insights from the dataset.

The tables and images presented in the study are of adequate quality, being clear, well-organized, and properly labeled.

Reviewer #5: Yes, the analysis presented match the analysis plan;

Yes, the results are clearly;

No, the manuscpript has more figures and tables than necessary.

**Conclusions**

-Are the conclusions supported by the data presented?

-Are the limitations of analysis clearly described?

-Do the authors discuss how these data can be helpful to advance our understanding of the topic under study?

-Is public health relevance addressed?

Reviewer #1: In the conclusions the authors detail the contributions of the study and the information and production undertaken to support the reactivation of TIPs, the access constraints created by the pandemic were well explained and the procedures used to address them.

Reviewer #4: The conclusions of the work align with the data presented but are limited to the analyses conducted. The authors touch on the key aspects of the data and its relevance, but they do not delve deeply into how the findings could contribute to a broader understanding of the topic. The study primarily focuses on descriptive analysis rather than using the data to expand our understanding of the subject at a deeper level. The study addresses public health relevance, particularly through the analysis of the Family Health Strategy and community engagement in health surveillance. The data highlight the importance of these elements in strengthening health interventions and improving the sustainability of health promotion initiatives.

Reviewer #5: Yes, the conclusions is supported by the data presented;

Yes, The limitations are described;

Yes, the authors do discuss how these data can be helpful to advance our understanding of the topic under study;

Yes, the public health relevance is addressed?

**Editorial and Data Presentation Modifications?**

Reviewer #1: No

Reviewer #4: Regional Health Offices (RHOs) and Regional Health Superintendence (RHS): What are the differences between them and which regions do they cover?

This sounds more like a figure legend than results:

The 337 number of TIPs installed in each municipality follows a similar color pattern described in Fig 3. Municipalities with one to five TIPs installed are highlighted in yellow, six to 10 in light orange, 11 to 15 in dark orange, and 16 to 19 in brown. Municipalities with no TIPs installed in that zone are shown in white.

Excessive figures, should be combined among them.

Participan, coordinators comments could be moved to supplementary material and not included in the main results of the study.

Reviewer #5: The presented manuscript describes information about the TIPS strategy for T. cruzi vector monitoring at the municipal level at Jequitinhonha Valley in the state of Minas Gerais, Brazil. This is a very interesting approach to evaluating how well the PNCDC is implemented among regions and proposing interventions based on scientific evidence. One of the best aspects of your manuscript is the discussion about PITs coverage, usage, and the disuse rate by establishment type.

The manuscript has merit for publication in PLOS NTD, however, a few revisions and corrections are required before acceptance.

General Comments

This manuscript brings to the scientific level very useful information to support public policies and propose interventions based on scientific evidence. Perhaps some aspects need to be better described, and some English proof reviews need to be done.

The manuscript is quite a bit longer than normally observed, and this is mainly due to the speech transcription included in the document. Please consider moving all the transcriptions to supplementary material. This way, the general readers interested will not be surprised by the number of your manuscript pages, and all the information will be easily available to the specific readers interested in the details on the transcriptions.

The presented manuscript is longer than normally observed. Please review and reduce most of the sections, especially the Results section.

Please review some of the acronyms used (eg. VC, PHU, RHS, etc...).

To consider making an English proof review of the document. For example, in the sentence:

“135 From this perspective, the community should direct any insects

136 suspected of being triatomines to the designated surveillance locations in their

137 countries: in Brazil, to the TIPs, as mentioned above.”

Consider: “From this perspective, the community should address any insects to the designated surveillance locations within their respective countries; in Brazil, for instance, to the TIPs, as previously mentioned.”

Abstract

I suggest you make the main objective clear in the abstract. When you use the sentence:

30 “For the first time in

31 scientific literature, the operational characterization, usage, and structuring of TIPs are

32 analyzed, alongside identifying factors hindering their sustainability.”

You are calling attention to the primacy of your study, and this is quite fine; perhaps, you do not inform us the main objective of your study clear at this section.

Author Summary

I suggest making an English proofread of this section. As an example, see the sentence above:

“55 In Brazil, when the community finds insects suspected of being triatomines, vectors

56 of the agent that causes Chagas Diseae (CD), they should take them to a Triatomine

57 Information Center (TIP), which can be in schools, basic health units, etc.”

Consider this revision:

“In Brazil, when the community finds insects suspected of being triatomines, the insects vectors of the agent that causes Chagas Disease (CD), they should take them to a Triatomine Information Center (TIP), which may be located in public infrastructure, like schools, basic health units, or other community facilities.”

Introduction

You have a very well-designed section, describing important information on the T. cruzi vector surveillance and the TIPs strategy of monitoring. Perhaps, try to be so concise in other sections of the manuscript.

Line 75: Chagas, 1909² to (Chagas, 1909)².

Materials and Methods

Specific Comments

Lines 66-71: "Their circadian activity and tolerance to human presence was used as a measure of establishment..." – Consider moving this information to the Methods section, as it seems out of place at the end of the Introduction.

Results

This section is fine and informative, perhaps once your manuscript is quite long, please consider reviewing and reduce this section, and then I can provide a detailed revision of the sentences.

Please consider moving the sentence above to the Introduction section.

“268 The implementation of TIPs in the health regions began in 1999 in the

269 Januária RHO, accounting for 26.4% of the total TIP installations in the study area.

270 Between 2000 and 2010, the process intensified in the Pedra Azul, Pirapora, and

271 Unaí RHOs regions, where 39.4% of the installations occurred. In the more recent

272 period, from 2011 to 2020, the Montes Claros RHS recorded 34.2% of TIP

273 installations in its municipalities.”

Please consider moving all the transcriptions to supplementary material.

Discussion

This section is fine and informative, perhaps once your manuscript is quite long, please consider reviewing and reduce this section, and then I can provide a detailed revision of the sentences.

Tables and Frames

All the tables need to be formatted as PNTD suggests (see https://journals.plos.org/plosntds/article/figure?id=10.1371/journal.pntd.0008359.t001).

Figures

You have an excessive number of figures. Please, consider grouping most of them into thematic figures.

**Summary and General Comments**

Reviewer #1: The study provides a reliable description of the operation, structure and use of TIPs in a highly endemic area of Chagas disease in Brazil since the 1990s. It is an important study because it evaluates, over a significant period of time, the entomological surveillance system that has been established as an effective tool in other countries. The successes and limitations that emerge over time will be of great value to

control programmes in the region. The experimental design is correct and the results are based on clear statistical analysis. The discussion has been developed on the basis of the interpretation of the results. The study falls within the scope of the journal, and adjustments were appropriately realized.I had suggested reducing the figures, but after the explanation given by the authors, I consider that the tables and figures should be included as proposed by them.

Reviewer #4: The study provides an analysis of TIPs using a mixed-methods approach, focusing on their physical, structural, situational, and spatial characteristics, as well as community engagement, the Family Health Strategy, and health professionals' perceptions regarding the sustainability of TIPs.

The study’s objectives are clearly defined, and appropriate methods are applied for data collection and analysis. Additionally, it addresses a relevant public health issue, particularly concerning community participation and the Family Health Strategy in improving health surveillance. The tables and figures are well-organized and contribute to the clarity of the presented results.

However, the data analysis is limited, as it primarily relies on descriptive statistics, which restricts the ability to explore the relationships between the studied factors in depth. The use of more advanced analytical techniques would provide a more nuanced view of the data. Moreover, the manuscript is excessively long, with a large number of figures that could be reduced or combined to avoid redundancy. The results and discussion sections could also be more concise, focusing on the most relevant findings and their impact.

Although the study highlights the relevance of TIPs for public health, it does not fully explore how its findings could contribute to advancing theoretical and practical knowledge in the field. A more extensive discussion of the implications of the findings for public policy and future research directions would strengthen the study's impact.

Overall, the study is well-executed in terms of data collection, but the lack of depth in analysis and the need to reduce the length and redundancy of the text limit its clarity and focus. Incorporating more complex analytical techniques and condensing the redundant sections would improve its academic rigor and practical relevance. Additionally, it would be helpful to include policy recommendations based on the findings and suggest directions for future research on TIP sustainability in different contexts.

Reviewer #5: The presented manuscript describes information about the TIPS strategy for T. cruzi vector monitoring at the municipal level at Jequitinhonha Valley in the state of Minas Gerais, Brazil. This is a very interesting approach to evaluating how well the PNCDC is implemented among regions and proposing interventions based on scientific evidence. One of the best aspects of your manuscript is the discussion about PITs coverage, usage, and the disuse rate by establishment type.

The manuscript has merit for publication in PLOS NTD, however, a few revisions and corrections are required before acceptance.

General Comments

This manuscript brings to the scientific level very useful information to support public policies and propose interventions based on scientific evidence. Perhaps some aspects need to be better described, and some English proof reviews need to be done.

The manuscript is quite a bit longer than normally observed, and this is mainly due to the speech transcription included in the document. Please consider moving all the transcriptions to supplementary material. This way, the general readers interested will not be surprised by the number of your manuscript pages, and all the information will be easily available to the specific readers interested in the details on the transcriptions.

The presented manuscript is longer than normally observed. Please review and reduce most of the sections, especially the Results section.

Please review some of the acronyms used (eg. VC, PHU, RHS, etc...).

To consider making an English proof review of the document. For example, in the sentence:

“135 From this perspective, the community should direct any insects

136 suspected of being triatomines to the designated surveillance locations in their

137 countries: in Brazil, to the TIPs, as mentioned above.”

Consider: “From this perspective, the community should address any insects to the designated surveillance locations within their respective countries; in Brazil, for instance, to the TIPs, as previously mentioned.”

Abstract

I suggest you make the main objective clear in the abstract. When you use the sentence:

30 “For the first time in

31 scientific literature, the operational characterization, usage, and structuring of TIPs are

32 analyzed, alongside identifying factors hindering their sustainability.”

You are calling attention to the primacy of your study, and this is quite fine; perhaps, you do not inform us the main objective of your study clear at this section.

Author Summary

I suggest making an English proofread of this section. As an example, see the sentence above:

“55 In Brazil, when the community finds insects suspected of being triatomines, vectors

56 of the agent that causes Chagas Diseae (CD), they should take them to a Triatomine

57 Information Center (TIP), which can be in schools, basic health units, etc.”

Consider this revision:

“In Brazil, when the community finds insects suspected of being triatomines, the insects vectors of the agent that causes Chagas Disease (CD), they should take them to a Triatomine Information Center (TIP), which may be located in public infrastructure, like schools, basic health units, or other community facilities.”

Introduction

You have a very well-designed section, describing important information on the T. cruzi vector surveillance and the TIPs strategy of monitoring. Perhaps, try to be so concise in other sections of the manuscript.

Line 75: Chagas, 1909² to (Chagas, 1909)².

Materials and Methods

Specific Comments

Lines 66-71: "Their circadian activity and tolerance to human presence was used as a measure of establishment..." – Consider moving this information to the Methods section, as it seems out of place at the end of the Introduction.

Results

This section is fine and informative, perhaps once your manuscript is quite long, please consider reviewing and reduce this section, and then I can provide a detailed revision of the sentences.

Please consider moving the sentence above to the Introduction section.

“268 The implementation of TIPs in the health regions began in 1999 in the

269 Januária RHO, accounting for 26.4% of the total TIP installations in the study area.

270 Between 2000 and 2010, the process intensified in the Pedra Azul, Pirapora, and

271 Unaí RHOs regions, where 39.4% of the installations occurred. In the more recent

272 period, from 2011 to 2020, the Montes Claros RHS recorded 34.2% of TIP

273 installations in its municipalities.”

Please consider moving all the transcriptions to supplementary material.

Discussion

This section is fine and informative, perhaps once your manuscript is quite long, please consider reviewing and reduce this section, and then I can provide a detailed revision of the sentences.

Tables and Frames

All the tables need to be formatted as PNTD suggests (see https://journals.plos.org/plosntds/article/figure?id=10.1371/journal.pntd.0008359.t001).

Figures

You have an excessive number of figures. Please, consider grouping most of them into thematic figures.

PLOS authors have the option to publish the peer review history of their article (what does this mean? ). If published, this will include your full peer review and any attached files.

**Do you want your identity to be public for this peer review?** For information about this choice, including consent withdrawal, please see our Privacy Policy .

Reviewer #1: No

Reviewer #4: No

Reviewer #5: No

**Figure resubmission:**
---

## [Editor Report · Decision Letter 2]

Dear Dra Ferreira,

We are pleased to inform you that your manuscript 'Community-based surveillance of Chagas Disease: characterization and use of triatomine information posts (TIPs) in a high-risk area for triatomine reinfestation in Latin America Triatomine information posts in an endemic area for Chagas disease' has been provisionally accepted for publication in PLOS Neglected Tropical Diseases.

Best regards,

Eric Dumonteil, Ph.D.

Academic Editor

Audrey Lenhart

Section Editor

Shaden Kamhawi

co-Editor-in-Chief

Paul Brindley

co-Editor-in-Chief

---

## [Editor Report · Acceptance letter]

Dear Dra Ferreira,

We are delighted to inform you that your manuscript, "Community-based surveillance of Chagas Disease: characterization and use of triatomine information posts (TIPs) in a high-risk area for triatomine reinfestation in Latin America Triatomine information posts in an endemic area for Chagas disease," has been formally accepted for publication in PLOS Neglected Tropical Diseases.

Best regards,

Shaden Kamhawi

co-Editor-in-Chief

Paul Brindley

co-Editor-in-Chief
